# INTERROGATING PARADIGMS IN SELF-SUPERVISED GRAPH REPRESENTATION LEARNING

## ABSTRACT

Graph contrastive learning (GCL) is a newly popular paradigm for self-supervised graph representation learning and offers an alternative to reconstruction-based methods. However, it is not well understood what conditions a task must satisfy such that a given paradigm is better suited. In this paper, we investigate the role of dataset properties and augmentation strategies on the success of GCL and reconstruction-based approaches. Using the recent population augmentation graph-based analysis of self-supervised learning, we show theoretically that the success of GCL with popular augmentations is bounded by the graph edit distance between different classes. Next, we introduce a synthetic data generation process that systematically controls the amount of style vs. content in each sample- i.e. information that is irrelevant vs. relevant to the downstream task- to elucidate how graph representation learning methods perform under different dataset conditions. We empirically show that reconstruction approaches perform better when the style vs. content ratio is low and GCL with popular augmentations benefits from moderate style. Our results provide a general, systematic framework for analyzing different graph representation learning methods and demonstrate when a given approach is expected to perform well.

## 1 INTRODUCTION

Analyzing graph-structured data is essential for many real-world applications and graph neural networks (GNNs) have emerged as a popular solution for challenging prediction tasks. However, these tasks often have limited labeled data due to prohibitive procuration costs and require models to possess strong generalization abilities to be practically useful. For example, in molecular property prediction, obtaining training labels requires expensive wet lab experiments and, at test time, models must predict properties for novel candidate molecules (Hwang et al., 2020; Duvenaud et al., 2015; Zitnik et al., 2018). Unsupervised graph representation learning is a natural paradigm in such cases, where graph contrastive learning (GCL) is a promising approach over using pre-training tasks (Hu et al., 2020).

While recent findings (Arora et al., 2019; HaoChen et al., 2021; Tian et al., 2020; von Kügelgen et al., 2021; Zimmermann et al., 2021; Wang & Isola, 2020; Purushwalkam & Gupta, 2020) have investigated what makes for successful *visual* contrastive learning, a similar understanding remains lacking for GCL. Furthermore, in computer vision, reconstruction-based approaches using an autoencoder framework (Kingma & Welling, 2014), updated with modern encoder architectures and data augmentation, are emerging as an alternative paradigm that avoids negative sampling or large batch sizes (Falcon et al., 2021). Reconstruction-based approaches (Kipf & Welling, 2016) have not yet been similarly revisited with improved GNN architecture designs or augmentations for self-supervised graph representation learning. Moreover, due to complications arising from the discrete, non-Euclidean nature of graph datasets, analysis from visual contrastive learning (VCL) cannot be straightforwardly extended to a graph setting. Therefore, it remains unclear under what circumstances a reconstruction or CL-based approach is expected to perform well on a given task.

In this paper, we investigate, theoretically and empirically, the conditions that enable a given unsupervised learning approach to perform well. Specifically, we first show theoretically that the success of GCL with generic graph augmentations (GGA) introduced by GraphCL (You et al., 2020a) is dependent on the graph edit distance between classes. Next, we address the elephant in the room:

that *untrained* GNNs have enough inductive bias to nullify the benefits of unsupervised pre-training on benchmark graph classification datasets. Therefore, we introduce a carefully designed synthetic dataset and conduct an extensive evaluation to better understand the behavior of both reconstruction and CL approaches. Here, we demonstrate that the effectiveness of different unsupervised approaches over strong untrained baselines can be understood through a style vs. content decomposition: the proportion of relevant information for a task (content) and irrelevant information (style) that each example in a dataset contains.

Our contributions are as follows: (i) We provide theoretical analysis of when contrastive learning is expected to work well, showing that this depends on the graph edit distance of samples within and across classes; (ii) We systematically evaluate reconstruction as an alternative unsupervised paradigm to contrastive learning for graphs, including introducing augmentation-augmented graph autoencoders (AAGAE). Further, we empirically show to what extent untrained GNN models are a competitive baseline in terms of accuracy, invariance, and sample complexity; and (iii)mWe identify a style vs. content trade-off in graphs and introduce an extensive benchmark setup that can carefully control this trade-off. Using our benchmark, we show how not only the ratio of style vs. content but also how generic versus content-aware augmentation impacts different learning paradigms.

## 2 BACKGROUND: UNSUPERVISED REPRESENTATION LEARNING FOR GNNS

While the general approach of pre-training can be either supervised (Hu et al., 2020)) or unsupervised, we focus on the latter. In this section, we formalize unsupervised representation learning (URL) for graphs and discuss two widely adopted approaches, namely graph contrastive learning and reconstruction-based learning. Appendix A.6 contains additional related work on self-supervised learning and data augmentation for graphs.

**Graph Contrastive Learning.** Contrastive learning (CL) frameworks learn representations by maximizing similarity between positive or augmented examples, and at the same time minimizing similarity between negative or uncorrelated examples. Formally, let $\overline{\mathcal{X}} = \{\overline{x}_1, \ldots, \overline{x}_n\}$ denote a dataset consisting of un-augmented (clean) samples, where each $\overline{x}_i = (G_i, F_i)$ corresponds to a tuple containing the adjacency matrix $G_i \in [0,1]^{n \times n}$ and node feature matrix $F_i$. Let $\mathcal{A}$ represent a set of augmentations over $\overline{\mathcal{X}}$; namely the generic graph augmentations (GGA) introduced by (You et al., 2020a): (i) node dropping, (ii) edge perturbation, (iii) attribute masking and (iv) sub-graph sampling. Furthermore, let $\mathcal{X}$ be the set of all augmented samples given $\overline{\mathcal{X}}$, and $\{x_i = \mathcal{A}(\overline{x}_i), x_j = \mathcal{A}(\overline{x}_i)\}$ be considered a positive pair.

GraphCL parallels SimCLR (Chen et al., 2020a) and uses the normalized temperature scaled cross entropy (NT-XENT) loss to learn representations by contrasting the representations of positive pairs and negative samples. Specifically, let $f$ be a graph feature learner, such that $f(x_i) = z_i \in \mathbb{R}^d$, e.g., a GNN with a global READOUT layer, *i.e.*, READOUT $: \mathbb{R}^{n \times d} \to \mathbb{R}^d$. Given batch size $B$, similarity function, sim$: (\mathbb{R}^d, \mathbb{R}^d) \to [0,1]$, temperature parameter, $\tau$, and positive pair, $\{x_i, x_j\}$, the NT-XENT loss is defined as:

$$\ell_{i,j} = -\log \frac{\exp\left(\text{sim}\left(z_i, z_j\right)/\tau\right)}{\sum_{k=1}^{2B} \mathbb{1}_{[k \neq i]} \exp\left(\text{sim}\left(z_i, z_k\right)/\tau\right)}. \tag{1}$$

Here, the numerator encourages the representations for $x_i, x_j$ to have high similarity, while the denominator encourages representations of negative pairs ($k \neq i$) to have low similarity. By maximizing similarity between positive samples, we expect the representations to become invariant to the properties modified by augmentations. Correspondingly, models learn to perform instance discrimination where each sample defines its own class and the augmented samples also belong to this class. Other recent GCL frameworks follow this general formulation but differ in the choice of the objective function and the augmentation strategy. For example, InfoGraph (Sun et al., 2020) maximizes the mutual information between sampled subgraphs (local) and pooled graph (global) representations. MVGRL (Hassani & Ahmadi, 2020) contrasts representations of graphs augmented through diffusion processes at node and graph scales. While this work focuses on GraphCL, other formulations including InfoGraph, DGI (Hjelm et al., 2019), and GMI (Peng et al., 2020), can be derived as instances of this general framework.

**Reconstruction-Based Approaches.** In computer vision, auto-encoding frameworks enriched with sophisticated model architectures and strong data augmentations (Falcon et al., 2021) are currently

being revisited as an alternative unsupervised learning paradigm that is less dependent on large batch-sizes (Chen et al., 2020a) and negative sampling strategies (Kalantidis et al., 2020) when compared to CL. Given that graph datasets are often significantly smaller than vision datasets and negative sampling strategies may be difficult to design, such approaches are particularly relevant to self-supervised graph representation learning and bear revisiting in light of stronger GNN architectures and augmentations.

Formally, let $g: (G_i, F_i) \rightarrow \mathbb{R}^{n \times d}$ be an encoder that outputs node representations, $h : \mathbb{R}^{n \times d} \rightarrow [0, 1]^{n \times n}$ be a decoder that predicts the edges of $G_i$ given node representations, and READOUT : $\mathbb{R}^{n \times d} \rightarrow \mathbb{R}^d$ provide graph representations given node representations. Then, a vanilla graph auto-encoder minimizes: $||h(g(x_i) - G_i||_2^2$, where $h$ is often defined as $\sigma\left(g(x_i)g(x_i)^T\right)$ and the aggregated graph representation $\boldsymbol{z}_i = \text{READOUT}(g(G_i, F_i))$ is used to perform downstream tasks. Given that graphs are generally sparse, standard implementations sample an equivalent number of positive and negative edges to ensure stable training. Variational graph auto-encoders (Kipf & Welling, 2016) reparameterize node representations and add a KL divergence term to the reconstruction loss similar to (Kingma & Welling, 2014). As we will argue in this paper, reconstruction-based approaches enhanced with suitable augmentations can be an effective alternative for GCL, under specific conditions. We choose to focus on reconstruction tasks as it is more general than task-specific pretraining tasks and more amenable to theoretical analysis (Khemakhem et al., 2020). In subsequent sections, we seek to understand when and why a given method performs well. We begin by taking a closer look at the performance of graph URL using benchmark datasets.

## 3   A Closer Look into Graph URL using Benchmark Datasets

Designing an unsupervised representation learning pipeline for graph-structured data requires selecting from a number of components, including encoder architecture, data augmentation, strategies for leveraging inductive bias, and training paradigms. It is important to understand the impact of different components on downstream performance, so practitioners can appropriately select them for their needs and determine if URL or pre-training will improve representation quality. To this end, we begin by performing an empirical study on benchmark datasets that considers different: (i) levels of inductive biases by incorporating data augmentation and varying the amount of training; (ii) representation learning paradigms (CL, reconstruction-based) (iii) GNN architectures (GIN, GCN, GAT etc.). We note that we are the first to implement and benchmark the augmentation augmented graph auto-encoders (AAGAE) as a stable reconstruction-based approach for graph URL.

While our empirical analysis offers several insights into different components of graph URL pipelines, we are unable to evaluate context-aware augmentations (CAA) as they are difficult to realize on standard graph benchmarks. CAA have been critical to the advancements in visual CL and the remainder of this paper investigates if CAA have similar promise for graph URL. In Section 4, we extend analytical tools from visual CL to characterize the behaviour of *generic* graph augmentations (GGA). In Section 5, we introduce a novel, synthetic benchmark that gives us control over the style vs. content ratio in the synthesized samples. This enables systematic evaluation of the potential gains to be obtained from CAA. We begin by evaluating the quality of representations obtained not only through GCL and reconstruction-based approaches but also from surprisingly effective, untrained GNN encoders.

### 3.1   Representation Learning Strategies

We consider three flavors of unsupervised representation learning: (i) *Graph contrastive learning*: For a representative GCL approach, we select GraphCL with GGA as it is a popular and effective method for graph classification. Following You et al. (2020a), we use random node-dropping or sub-graph sampling at $20\%$ of the graph size as the augmentation strategy; (ii) *Reconstruction-based*: In addition to standard graph autoencoders (GAEs), we extend AAVAE (Falcon et al., 2021) to the graph domain and introduce augmentation-augmented graph autoencoders (AAGAE). While GAE minimizes the reconstruction loss with respect to the original sample, AAGAE minimizes the reconstruction loss between the original and the reconstruction for an augmented sample to learn representations that are consistent with respect to augmentations: $||G_i - h(g(A(G_i, X_i)))||_2^2$. We use the same augmentations and encoder architecture as GraphCL for fair comparison and include a straight-through estimator in the decoder for better training (Jang et al., 2017); and finally (iii) *Untrained*

*representations*: We include untrained GNNs as an important baseline and find that representations from randomly initialized models often perform comparably to unsupervised approaches.

There is some folk wisdom that untrained GNNs can be surprisingly competitive with trained models. Early GNN works noted that even an untrained model may have strong inductive bias suitable for node-level tasks (Kipf & Welling, 2017). For graph-level transfer learning, Xu et al. (2021) consider an untrained GIN model as a baseline, finding it obtains competitive although generally inferior results. A recent blog post performs a limited exploratory analysis of graph classification using untrained embeddings obtained from a simple GCN model (Safronov, 2021). Our analysis here (extended to consider different architectures in Appendix A.3-A.4) is far more comprehensive, showing precisely in what ways untrained models may be competitive.

Table 1: **Inductive Bias on Benchmark Datasets.** We report the performance of *un*trained N-layer GIN encoders against 3-layer encoders trained through GraphCL (You et al., 2020a) or reconstruction approaches (GAE/AAGAE). Results for GraphCL are taken from the paper and we follow the same evaluation protocol. Best results are indicated in bold; results within standard deviation are underlined.

| Dataset | Untrained (3) | Untrained(4) | Untrained (5) | GraphCL | GAE | AAGAE |
|---|---|---|---|---|---|---|
| MUTAG (188) | $85.76 \pm 7.38$ | $86.36 \pm 6.51$ | $86.73 \pm 10.33$ | $86.80 \pm 1.34$ | $87.76 \pm 3.00$ | $\mathbf{88.23 \pm 0.98}$ |
| PROTEINS (1113) | $73.64 \pm 5.464$ | $74.46 \pm 4.09$ | $74.22 \pm 2.85$ | $74.39 \pm 0.45$ | $\mathbf{75.36 \pm 0.4}$ | $74.77 \pm 0.43$ |
| NCI1 (4060) | $70.65 \pm 1.99$ | $70.36 \pm 3.11$ | $70.49 \pm 2.42$ | $77.81 \pm 0.41$ | $79.48 \pm 0.44$ | $\mathbf{79.75 \pm 1.25}$ |
| DD (1187) | $73.23 \pm 8.25$ | $72.15 \pm 7.25$ | $77.08 \pm 4.18$ | $\mathbf{78.62 \pm 0.40}$ | $78.24 \pm 0.67$ | $77.59 \pm 0.64$ |
| RDT-B (2000) | $72.34 \pm 6.64$ | $64.57 \pm 8.03$ | $67.32 \pm 7.41$ | $\mathbf{89.53 \pm 0.84}$ | $79.75 \pm 1.25$ | $79.95 \pm 4.39$ |
| IMDB-B (1000) | $67.22 \pm 7.77$ | $61.26 \pm 7.01$ | $60.43 \pm 5.92$ | $71.14 \pm 0.44$ | $\mathbf{71.70 \pm 0.36}$ | $71.26 \pm 0.305$ |

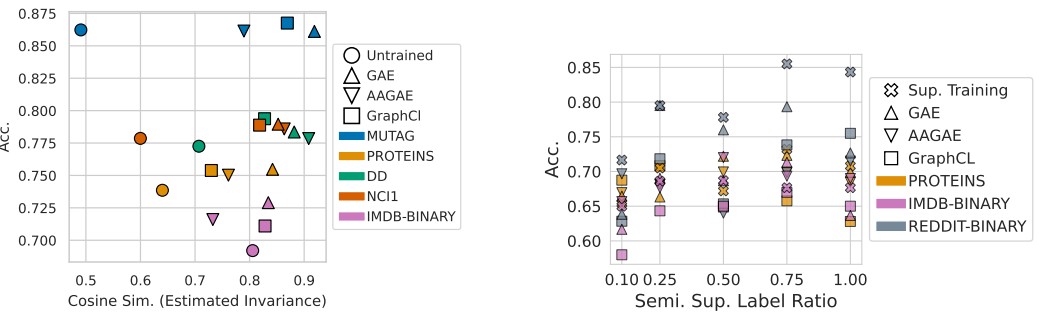

(a)  (b)

Figure 1: (a) **Invariance vs. Acc on TUDatasets.** While trained models do see an improvement in both accuracy and invariance, we see that this improvement in often minimal over an untrained model, suggesting that GGA augmentations introduce invariance that is not always useful to the downstream tasks; (b) **Sample Complexity on TU Datasets.** We perform semi-supervised learning on TU Datasets with various amounts of labeled data. For REDDIT-BINARY, pretraining does not improve sample complexity. GAE/AAGAE have slightly better complexity than the supervised baseline on IMDB-BINARY. On PROTEINS, pretraining also does not have clear benefits.

## 3.2 EVALUATION METRICS

While prediction accuracy on the downstream task is an important criterion for choosing a representation learner, we consider two other metrics to empirically illustrate the benefits of these methods, or lack thereof, over the untrained representations. Specifically, we quantify how training with unsupervised methods affects the invariance of representations with respect to augmentations, and sample complexity. These metrics are selected because they are well-aligned with the objectives of potential applications. For example, in molecular property prediction, it is desirable to have models that are invariant to small measurement discrepancies and can quickly generalize to new distributions. We define these measures and describe how they are computed as follows.

*Invariance:* When training with NT-XENT, as GraphCL does, the representation similarity between augmented pairs is maximized and models learn invariances to augmentations. AAGAE also enforces consistency to natural and augmented samples. However, it is unclear if (i) models are actually learning invariant representations and (ii) if this property is valuable to downstream task performance. To measure this property, for a given sample $\bar{x}_i$, we generate 300 randomly augmented samples $\{x_i^1 \ldots x_i^{300}\}$ compute the average cosine similarity between $\bar{x}_i, x_i^j$. Note, using cosine similarity as a proxy for the invariance property is well-motivated by NX-ENT.

*Sample Complexity:* If unsupervised learning is used as the pre-training step, then it is expected that the pre-trained model will have improved sample complexity. We consider if this holds true for GraphCL, GAE and AAGAE. Specifically, after unsupervised training, we include a linear classifier layer to the backbone or encoder. Then, we train the model, end-to-end, in a supervised fashion, where we vary the size of the labeled dataset, extending the setup of (You et al., 2020a).

**GGA introduces invariance that has limited improvements in performance.** Fig. 1a plots the invariance of GraphCL, GAE, AAGAE, and untrained models on benchmark TU datasets. As noted in Tab. 1, untrained models often perform comparably to trained ones ($\leq 5\%$ difference). However, we see that training with GCL or reconstruction leads to improved invariance. This suggests that the learning invariance was not necessary for downstream task performance. While designing better augmentations is a difficult but obvious solution (Zhao et al., 2020; Kong et al., 2020), in Sec. 5, we find that even optimal augmentations are often unable to surpass supervised performance and discuss implications of this result.

**Pretraining has limited benefits on sample complexity.** Given that GCL or reconstruction did not lead to considerable improvements in accuracy over a untrained baseline or introduce meaningful invariance, we investigate if pretraining leads to better sample complexity. As shown in Fig. 1b, we find there is no clear benefit to pretraining. On REDDIT-BINARY, the fully supervised model has better accuracy across all label ratios. For IMDB-BINARY, reconstruction based approaches offers a slight improvement. The results are mixed for PROTEINS. We note that the supervised model did not see consistent improvements as the labeled ratio increased and suspect this is related to the random data-splits (Dwivedi et al., 2020).

Our empirical analysis demonstrates the unreasonable effectiveness of untrained GNNs and highlights the uneven benefits GGA with respect to invariance, sample complexity and accuracy. Moreover, in the appendix, we show that URL leads to minimal improvements for expressive architectures, such as PNA (Corso et al., 2020). In the next section, we use the recently proposed Population Augmentation Graph (HaoChen et al., 2021) to better understand the limitations of GGA, as they are routinely used when it is unclear how to leverage domain knowledge as graph augmentations.

## 4 USING POPULATION AUGMENTATION GRAPHS TO ANALYZE GGA

Recent attempts to analyze theoretically the performance of contrastive learning often assumes that sample views are independent, a condition clearly violated by data augmentation (Arora et al., 2019; Tosh et al., 2021). To avoid this assumption, HaoChen et al. (2021) recently introduced the notion of a *population augmentation graph*, which represents augmented samples as nodes and edges as the likelihood of generating a given pair of augmented samples from the same clean sample. HaoChen et al. (2021) also provably showed that well-designed augmentations will lead to tight subgraphs (partitions) in the population graph that correspond with downstream class labels. They show that true class labels can be recovered, up to some error, by performing spectral decomposition on this graph and propose a matrix factorization based objective function to perform the decomposition.

In this section, we leverage population augmentation graphs (PAGs) for analyzing contrastive learning with graph data, and show that employing generic graph augmentations from (You et al., 2020a) will lead to subgraphs in the PAG that are dependent on the graph edit distance (GED) between samples of different classes: that is, the minimum number of graph edit operators (operations such as node dropping or edge perturbation that also form the basis for the augmentations commonly used in contrastive learning) needed to transform one sample into another. This implies that performing spectral decomposition on this graph will not align with true class labels if the average GED of samples between two different classes is smaller than the average GED of samples between the same class. Equivalently, generic graph augmentation (GGA) can be interpreted as imposing GED as the implicit metric for representation similarity, and the success of GCL with GGA is dependent on how appropriate GED is for a given task.

More specifically, our analysis is comprised of the following steps: (i) defining the population augmentation graph; (ii) showing that GGA can be decomposed in graph edit operators; (iii) describing graph characteristics of the PAG; and finally (iv) showing that the resulting subgraphs of the PAG are determined by GED, which may not necessarily align with downstream class labels. We follow the notation in (HaoChen et al., 2021).

## 4.1 Constructing the PAG with Generic Graph Augmentations

Given a natural dataset $\overline{\mathcal{X}}$, let $\mathcal{A}(\cdot|\overline{\boldsymbol{x}}_i)$ be the distribution of augmentations given a natural sample $\overline{x}$, or, intuitively, as the probability of generating a particular augmented sample from the large but finite set of all possible augmented versions of $\overline{x}$. Then, $\mathcal{X} := \cup_{\overline{x} \in \overline{\mathcal{X}}} \mathcal{A}(\cdot|\overline{x})$. Note, the tuple $(\boldsymbol{x} = \mathcal{A}(\overline{x}), \boldsymbol{x}' = \mathcal{A}(\overline{x}))$ is considered a positive pair.

*Population Augmentation Graph* HaoChen et al. (2021): Let $\mathcal{G}^p$ be the population graph, where all $N$ samples in augmented set $\mathcal{X}$ form the nodes and $W \in \mathbb{R}^{N \times N}$ is the corresponding adjacency matrix. The edge weight between two nodes $x$ and $x'$ is defined as

$$w_{x,x'} := \mathbb{E}_{\overline{x} \in \mathcal{P}_{\overline{\mathcal{X}}}}[\mathcal{A}(x|\overline{x})\mathcal{A}(x'|\overline{x})].$$

Intuitively, if $w_{x,x'}$ is larger, it is relatively easier to generate the augmented pair from the same natural sample.

We first observe that augmentations defined by GraphCL (such as node dropping, edge perturbation, and sub-graph sampling) can be directly decomposed using standard graph edit operators (node dropping, node addition, edge dropping, and edge addition). For example, an edge perturbation augmentation can be represented as a set of edge drops and edge additions between $\{x, \mathcal{A}(x)\}$. Further, note that GraphCL assumes that constraining augmentations to a fraction, $\gamma$, of the overall graph size will preserve task-relevant information, and only one augmentation may be applied to a graph at a time. Then, for a single sample, $\overline{x}_i = (G_i, F_i)$, the maximum edit distance of an augmented sample $x' = \mathcal{A}(\overline{x})$ is:

$$\max\{\gamma|\mathcal{E}_i|, \gamma|\mathcal{V}_i|\},$$

where $\mathcal{E}_i, \mathcal{V}_i$ are the edge/node sets. Now, since graphs are discrete, the augmentation severity is restricted and only one edit can be applied at a time, we can completely delineate the set of allowable augmentations given $\overline{x}_i$, $\tilde{\mathcal{A}}_i$, and can determine the size of this set.

For example, consider the node dropping augmentation and let $\gamma_{\mathcal{V}_i} \in \mathbb{Z}^+$ be the number of allowable node drops for $G_i$. Then, there are

$$ND_i := \sum_{j=1}^{\gamma_{\mathcal{V}_i}} \frac{|\mathcal{V}_i|!}{(|\mathcal{V}_i| - j)!j!}$$

allowable augmented samples. Further, note that augmentations are applied randomly, so any sample in the augmentation set is equally likely, up to node permutation. This means, we can exactly define $\mathcal{A}(x|\overline{x})$, as $\frac{1}{|\tilde{\mathcal{A}}_i|}$. A similar analysis can be used for other augmentations. For ease of exposition, we focus on one augmentation here.

Then, we observe

$$\mathcal{A}(x|\overline{x}) \neq 0 \iff x \in \tilde{A}_{\overline{x}} \tag{2}$$
$$GED(x, \overline{x}) \leq \gamma_{\overline{x}}. \tag{3}$$

Intuitively, $x$ must exist in the augmentation set of $\overline{x}$ to have non-zero probability and the augmentation set is defined by all graphs within $\gamma_{\overline{x}}$ edits. This implies $w_{x,x'}$ will be large when $x, x'$ jointly appear in the augmentation set of many different natural samples. As such, we have defined all possible nodes in $\mathcal{G}^p$ as well as how the edges are defined and can discuss implications of the structure of the population graph induced by GGA.

## 4.2 Analyzing Population Augmentation Graph Structure

HaoChen et al. (2021) show that performing spectral decomposition on the augmentation graph provably recovers underlying classes if the partitions or subgraphs in the augmentation graph can be aligned to downstream classes. We now show how these partitions are related to GED and how the structure of these partitions relates to task performance. Clearly, for strong downstream performance, partitions must contain more samples from the same class than of different classes.

Recall Eq. 2 and Eq. 3 define when a given edge will exist between samples. Now, clearly, determining how many edges will cross between ground-truth classes can be determined by measuring the edit distance between samples of different classes. Formally, consider a simple binary task, where $\Psi := \{(GED(\overline{x_i}, \overline{x_j}) \leq \gamma_i + \gamma_j \ni y(\overline{x_i}) \neq y(\overline{x_j})\}$ corresponds to a set of edges across classes and $\Phi := \{(GED(\overline{x_i}, \overline{x_j}) \leq \gamma_i + \gamma_j \ni y(\overline{x_i}) = y(\overline{x_j})\}$ is the set of edges within classes. Then,

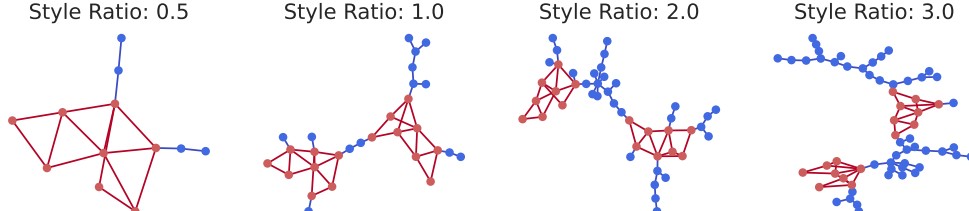

Figure 2: **Synthetic Dataset Generation.** We create a six class, graph classification dataset by first, selecting one of six unique motifs and then injecting 1-3x copies into a randomly generated tree. Here, the motifs completely determine the class, and are considered "content" (shown in red). To vary the amount of style, the size of the background tree graph (shown in blue) is ratio of the number of "content" nodes. Above are examples of varying style ratios. Our dataset goes beyond binary, benchmark datasets and allows for content-aware augmentations, a critical component to understanding the maximal performance GCL frameworks.

if $|\Psi| > |\Phi|$, it is not possible to learn well-aligned partitions because partitions will contain a mix of classes. Therefore, if spectral contrastive learning is performed using GGA, there is some fixed error, $\epsilon$, that is directly determined by the average GED between classes, as shown.

While our analysis leverages PAG and the spectral contrastive loss, HaoChen et al. (2021) show that their proposed loss function can be extended to other popular contrastive loss functions and also empirically performs well. Therefore, our results suggest generally that the success of graph contrastive learning with GGA is dependent on the GED between classes. The augmentation set of a given sample is determined by its graph size, and thus it should be more difficult to achieve invariance to GGA augmentations for larger graphs. Third, the above analysis may offer a path to deriving bounds on the generalization to certain distribution shifts, for example graph size shifts.

## 5 A STYLE VS. CONTENT PERSPECTIVE FOR EVALUATING CONTENT-AWARE AUGMENTATIONS

While the above analysis identifies untrained networks as a strong baseline for URL and characterizes GGA's behavior, it remains unclear how to improve different components of GCL and reconstruction-based approaches such that the additional expense of pretraining is justified. Given that existing augmentations are far from optimal (see Sec. 4), one avenue of improvement is through better augmentation design. However, graph data augmentation is known to be difficult (Kong et al., 2020; Zhao et al., 2020) and it is unclear if the different paradigms will benefit equally from improved augmentations. To that end, we first take the perspective that graph samples can be decomposed into style (*ir*relevant information) vs. content (task relevant information). We then introduce a synthetic data generation process that controls the amount of style vs. content in dataset samples (see Fig. 2), and allows for oracle, content-aware augmentations (CAA). Using this dataset, we validate the effectiveness of augmentations and introduce a valuable benchmark to the community.

Recent theoretical works in VCL also build upon a similar style vs. content perspective. For example, von Kügelgen et al. (2021) introduced a latent variable model to show that self-supervised training with data augmentations is able to recover a style vs. content partition in the latent representation and Zimmermann et al. (2021) showed that is possible to invert the data generating process using contrastive learning and augmentations. Our work differs in that our data generation process operates directly in the observed domain, instead of latent variables as in (von Kügelgen et al., 2021), and we leverage oracle augmentations to better understand the performance of different frameworks. **Synthetic Data Generation:** The proposed data generation process consists of three components: a set of $C$ motifs, $\mathcal{M}$, that uniquely determine $C$ classes, a random graph generator, $RBG(n)$, parameterized by the number of nodes (we can also equivalently define this based on number of edges), and $\rho$, the style multiplier, which controls how much irrelevant information is included in a sample. To generate a sample, we attach a randomly generated background graph (*i.e.*, style component) to a motif (*i.e.*, content) in accordance to the style multiplier. While this process is simple, it addresses several limitations often encountered in GCL evaluation. Specifically, this process (i) allows for varying levels of content-aware augmentation (*i.e.*, edges that can be perturbed directly in the background graph without harming the motif); (ii) is easily extended beyond binary classification; (iii) contains relatively large number of samples and (iv) offers a natural test bed for GNN size generalization or transfer learning.

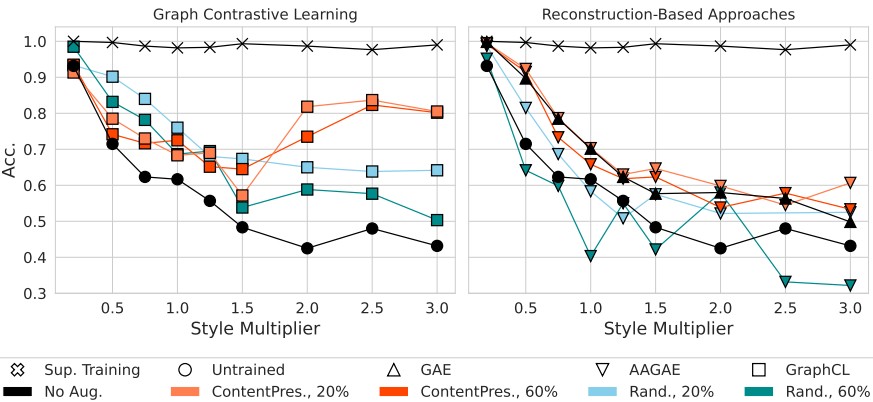

Figure 3: **Effect of Style vs. Content Ratio on Different Paradigms:** We evaluate the performance of contrastive and reconstruction approaches with content preserving and random augmentations as the ratio of style vs. content changes. As expected, reconstruction methods perform best in low style regimes. While we see that content aware augmentations improve performance, especially for GCL, in high style regimes, it remains difficult to match the supervised performance.

**Experimental-Setup:** We let $C = 6$ and define $RBG(n)$ through a random tree generator, where $n$ is $\rho\%$ of the nodes belonging the motif. To make the task more challenging, we randomly insert between 1-3 motif copies into each sample. We explore two different augmentation strategies, namely *random* edge dropping and *content preserving* edge dropping, where only edges in the background graph are dropped. We parameterize these augmentations by the number of edges in the total entire and background graphs, respectively. We do not consider any additional node features, though that extension would be straightforward. We use a 5-layer GIN-based encoder for both our GraphCL backbone and GAE/AAGAE encoders and the models are evaluated using a linear probing protocol (Chen et al., 2020a). See appendix A.2 for more details.

## 5.1 BALANCING STYLE VS. CONTENT

Many real graph datasets can be understood through the aforementioned partitioning of relevant vs. irrelevant information. For example, molecules can be split into functional groups (content) and carbon rings (style). A natural question is whether a reconstruction-based approach or contrastive learning approach is preferable if there is some intuition on the ratio of style vs. content (*i.e.*, many carbon rings vs. a few). We make the following observations from Fig. 3.

**Reconstruction-based approaches perform well in low style regimes.** Note that as the amount of style increases, the problem inherently becomes harder. However, in regimes where there is little style content (irrelevant information), reconstruction-based approaches should perform well in particular as the model will only learn to recover the relevant information. Conversely, GCL with GGA risks corrupting content, and generating false positives or out of distribution samples. Thus, we expect the accuracy of reconstruction-based approaches to fall more sharply as the amount of style increases, which indeed we do see in Fig. 3. We also find that that CAA helps GCL perform considerably better than with GGA, and that less severe generic augmentations are better; this supports the hypothesis that GCL succeeds when, whether by augmentation design or simply the composition of the data, there is less risk of the augmentations corrupting content.

**Content-aware augmentations are not a silver bullet.** Optimal augmentations generate views that only share task-relevant information and minimize other task-irrelevant information (Tian et al., 2020). For the proposed dataset, this would correspond to removing the entire background graph. We restrict augmentations to removing $60\%$ of the style portions and find even with strong content-preserving augmentations, unsupervised approaches significantly under-perform when compared to fully supervised models. This is in stark contrast to VCL where strong data augmentation is a critical component to surpassing performance of supervised models on a variety of vision tasks (Chen et al., 2020a). Our analysis suggests that other framework components, such as more expressive architectures (Chen et al., 2020b; Xu et al., 2019; Velickovic et al., 2018; Corso et al., 2020; Hamilton et al., 2017) and sampling strategies (Kalantidis et al., 2020; Grill et al., 2020; Chen & He, 2020), must also be developed before GCL sees the same success. Furthermore, we note that the gain from CAA in high-style regimes is much less pronounced for reconstruction approaches than for

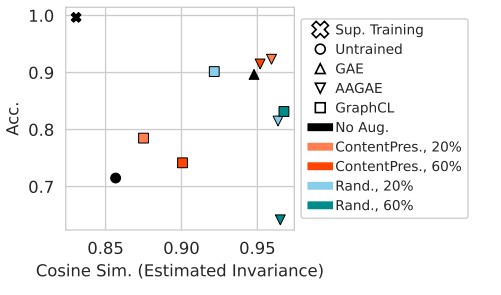 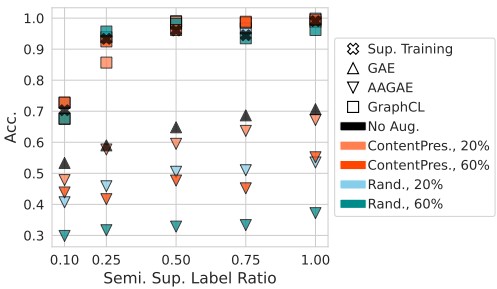

(a)                                     (b)

Figure 4: (a) **Invariance vs. Accuracy on Synthetic Datasets.** We measure the invariance to random augmentations, when $\rho = 0.5$; (b) **Sample Complexity on Synthetic Dataset.** We perform semi-supervised learning when $\rho = 1.0$ with various amounts of labeled data. GraphCL has considerably better performance than reconstruction-based methods. Content-aware augmentations are useful for both paradigms.

GCL. This may partially be attributed to increased difficulty in reconstructing larger graphs. More sophisticated decoders and algorithms may help improve performance.

## 5.2 EVALUATION

*Invariance:* Invariance to random edge dropping at 20% of the graph after performing unsupervised training when $\rho = 0.5$ is shown in Fig. 4a. Almost all methods improve on the untrained baselines in accuracy and invariance. We find that all reconstruction-based methods have high invariance in this low style regime. In contrast, GCL models that are more invariant tend to be more accurate, which may suggest that for this learning paradigm, learning to be invariant to augmentations can improve accuracy. Interestingly, supervised training is the most accurate but the least invariant–this reminds us the pursuit of expressive GNN models that most successfully map distinct graphs to distinct representations Xu et al. (2019) is helpful in supervised settings, though without the guidance of task supervision, unsupervised learning may benefit from greater representation invariance.

*Sample Complexity:* We measure the performance of different methods when $\rho = 1$ and at different labeling ratios in a semi-supervised setting. While CAA improves the sample efficiency of reconstruction-based methods with respect to GGA, we see that (i) GAE outperforms AAGAE across all labeling rates, and (ii) all considered reconstruction-based methods are not able to match the performance of GCL or supervised performance. This suggests that AAGAE may converge to a more unstable loss landscape, but also highlights our observation at the end of Sec. 5.1 that augmentations are limited in what they can accomplish, and may require technical advances such as more sophisticated decoders to come into their own to the extent they have in computer vision.

## 6 CONCLUSION

In this paper, we seek to understand the behavior of GCL and reconstruction-based approaches when performing unsupervised graph representation learning. We show theoretically that the success of GCL with popular, generic graph augmentations is directly dependent on the average graph edit distance between classes. Our empirical study shows in detail the competitiveness of untrained GNNs on benchmark datasets to help the community understand what benefits to expect from training. We demonstrate for both GCL and reconstruction-based methods, which we consider in greater detail for unsupervised graph representation learning including with a new method AAGAE, that generic graph augmentation does not introduce meaningful invariance or reliably improve model sample complexity on these benchmarks. Thus, we introduce a synthetic benchmark with a controlled style vs. content decomposition to understand how much can be gained from optimal (content-aware) augmentations and better compare the GCL and reconstruction paradigms. Our work provides useful frameworks and evaluation tools to better understand the performance of unsupervised graph representation learning, contributions we expect to be of great interest to this burgeoning field.

## 7 REPRODUCIBILITY STATEMENT

For reproducibility, we provide details about data generation in appendix A.2. For baselines, we build off the code provided by the authors and follow the authors' guidelines to set the hyperparameters. We will also release code at this link: https://anonymous.4open.science/r/GCLRecon-5DF6/

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

# A   APPENDIX

## A.1   UNDERSTANDING GENERIC GRAPH AUGMENTATIONS

We expand our discussion on the connections between generic graph augmentations and graph edit distance. We also discuss how the graph edit distance between dataset samples influences the structure of the population augmentation graph HaoChen et al. (2021), a recently introduced tool to understand contrastive learning.

Table 2: Notation

| Symbol | Definition |
|---|---|
| $\overline{\mathcal{X}}$ | the original or natural dataset. |
| $\mathcal{X}$ | set of all augmented data. |
| $x_i$ | data sample containing graph and node feature tuple, $(G_i, F_i)$ |
| $\mathcal{E}_i$ | Edge set of $G_i$. |
| $\mathcal{V}_i$ | Node set of $G_i$. |
| $\gamma \in (0, 1)$ | augmentation strength. Controls the % of edges or nodes that may be perturbed by the selected augmentation |
| $\mathcal{A}(\overline{x})$ | Augmentation, $\mathcal{A}$, applied to the natural sample $\overline{x}$ |
| $\mathcal{A}(\cdot|\overline{\boldsymbol{x}}_i)$ | distribution of augmentations given a natural sample, $\overline{\boldsymbol{x}}_i$. |
| $\mathcal{B}(\overline{x})$ | set of allowable augmentations given $\overline{x}$. |
| $\mathbf{D} \in \mathbb{Z}^{|\overline{\mathcal{X}}| \times |\overline{\mathcal{X}}|}$ | positive, symmetric distance matrix where $\mathbf{D}_{i,j} = GED(\overline{x_i}, \overline{x_j})$ |
| $\Gamma \in \mathbb{Z}^{1 \times |\overline{\mathcal{X}}|}$ | column vector containing max. number of allowable perturbations per sample. |
| $\mathbf{y} \in [0, 1]^{|\overline{\mathcal{X}}|}$ | column vector providing the labels of all natural samples. |

## A.1.1   GGA AND GRAPH EDIT DISTANCE

Graph edit distance ($GED$) is used to capture similarity between two graphs. Intuitively, it captures the cost of making elementary edit operations on a graph, $g_1$, to transform it to be isomorphic to another graph, $g_2$. Given two graphs, $g_1, g_2$,

$$GED(g_1, g_2) = \min_{(e_1, \ldots, e_k) \in \mathcal{P}(g_1, g_2)} \sum_{i=1}^{k} c(e_i),$$

where $\mathcal{P}(g_1, g_2)$ is the set of paths (series of edit operations) that transforms $g_1$ into $g_2$, $e_i$ is $i$-th edit operation in the path, and $c(e_i) > 0$ is the cost of the particular edit. In this work, we consider *node insertion*, *node deletion*, *edge deletion* and *edge addition* as the elementary graph edit operators as these are well-aligned to the augmentations defined in You et al. (2020a), namely *node dropping*, *edge perturbation*, *attribute masking*, and *sub-graph sampling*. While GED is typically defined on graph structure, our analysis can be extended to include categorical node attributes by introducing a graph operator that performs a "replacement" whenever a graph's node attributes disagree. Then, the GED is the cost of structural changes and the number of disagreements between their attributes. Categorical variables are common in molecular classification tasks, where attributes correspond to elements, and discrete node attributes are often used when analyzing GNNs (Xu et al., 2019). We also consider a constant cost of 1 per operation, such that GED counts the number of operations required to transform one graph into another.

For example, let $(g, g_a)$ represent the original and augmented graph respectively, where we perform *node dropping* to obtain $g_a$. Recall that the *node dropping* augmentation may only drop up to some fraction of nodes in $g$. Then, clearly the minimum cost path can then be found using only *node deletion* operators, and the $GED(g, g_a)$ is bounded by the number of allowed node drops. Similarly, if $g_a$ was obtained through the *edge perturbation* augmentation, which randomly adds or removes a fraction of edges, then $GED(g, g_a)$ is bounded by the number of allowable edge modifications and can be obtained using only *edge addition/deletion* operators. (Here, we allow nodes without edges to still exist, so performing node addition/deletion would not result in a lesser *GED*.) The *sub-graph sampling* augmentation extracts a connected sub-graph that contains at most a fraction of total nodes. The minimum cost path can then be defined using only *node deletions*, e.g. where the

operator is applied to all nodes not in the sampled sub-graph. Therefore, $GED(g, g_a)$ is bounded by $|g| - |g_a|$.

Given the aforementioned discussion, we can now define the set of allowable augmentations using *GED* and make the following remarks. Please see Table 2 for a complete list of notation.

**Definition 1** (Set of Allowable Augmentations). *Let $\mathcal{A}$ be a generic graph augmentation (node dropping, etc). Then, all allowable augmented samples induced by $\mathcal{A}(x_i)$ have graph edit distance less than $\max\{\gamma|\mathcal{V}_i|, \gamma|\mathcal{E}_i|\}$ to $x_i$. Equivalently:*

$$\mathcal{B}(x_i) \triangleq \{x' : GED(x', x_i) \leq \max\{\gamma|\mathcal{V}_i|, \gamma|\mathcal{E}_i|\}\}.$$

**Remark A.1** (Upper-bound on Size of Augmentation Set). *The size of $B(\overline{x_i})$ can be upper-bounded through a combinatorial or counting process. For example, to determine $B(\overline{x_i})$ when the considered augmentation is node dropping, we can delineate all sets of possible nodes with size upto $\gamma|\mathcal{V}_i|$. Formally, the upper-bound on the number of samples generated using node dropping are:*

$$|\mathcal{B}(\overline{x_i})| \leq \sum_{j=1}^{\gamma_{\mathcal{V}_i}} \frac{|\mathcal{V}_i|!}{(|\mathcal{V}_i| - j)! j!}$$

*We note that this value is an upper-bound because isomorphic pairs are treated as two separate graphs. Furthermore, note the size of the augmentation set grows exponentially with graph size.*

**Definition 2** (Overlapping Sample). *An augmented sample, $x'$, is considered an overlapping sample if belong to the augmentation set of multiple natural samples: $x' \in \mathcal{B}(\overline{x_i}) \wedge x' \in \mathcal{B}(\overline{x_j})$, where $i \neq j$.*

Using Def. 2, we show that overlapping examples must exist given certain conditions on graph edit distance of samples in $\overline{\mathcal{X}}$.

**Remark A.2** (Existence of Overlapping Samples). *Consider two samples $\overline{x_i}$ and $\overline{x_j}$. Let $r_i = \max\{\gamma|\mathcal{V}_i|, \gamma|\mathcal{E}_i|\}$ and $r_j = \max\{\gamma|\mathcal{V}_j|, \gamma|\mathcal{E}_j|\}$. If $GED(\overline{x_i}, \overline{x_j}) < r_i + r_j$, then $\exists x' \in B(\overline{x_i}) \wedge x' \in B(\overline{x_j})$, i.e. at least one augmented sample belongs to both the induced augmentation sets.*

**Definition 3** (Invalid Augmented Samples). *We consider an augmented sample, $x$ to be an invalid sample, if $x \in \mathcal{B}(\overline{x_i}) \wedge x \in \mathcal{B}(\overline{x_j})$ (an overlapping sampling), and $\mathbf{y}_i \neq \mathbf{y}_j$.*

**Claim A.1.** *Given $\mathbf{D}, \Gamma, \mathbf{y}$, we can lower-bound the number of overlapping samples in the empirical data distribution as $\frac{1}{2} \sum_{i,j \in [1,...,|\mathcal{X}|]} \mathbb{1}(\mathbf{D}_{ij} - \Gamma_i - \Gamma_j \leq 0)$ where $\mathbb{1}$ is the indicator function. Furthermore, if we consider oracle label information, we can lower bound the number of invalid samples as $\frac{1}{2} \sum_{i,j \in [1,...,|\mathcal{X}|]} \mathbb{1}((\mathbf{D}_{ij} - \Gamma_i - \Gamma_j)|\mathbf{y}_i - \mathbf{y}_j| < 0)$.*

*Proof.* $\Gamma_i + \Gamma_j$ is the total number of edit operations that can be applied to either samples $x_i$ or $x_j$. If the graph edit distance between samples $i$ and $j$ is smaller than this, then it is possible to reach the same augmented sample somewhere on the edit path that turns $x_i$ into $x_j$ regardless of which endpoint we start from. This augmented sample constitutes an overlapping sample, or an invalid sample if the class labels of $x_i$ and $x_j$ differ. Note that there may be multiple such augmented samples that can be created from either $x_i$ or $x_j$; our indicator function counts one per pair of samples, and thus helps constitute a lower bound. $\square$

### A.1.2 DISCUSSION ON INVALID SAMPLES

An invalid sample does not have a clear label because we do not know which natural label should be assigned to it. This can incur instability in discriminative methods if the invalid sample's loss is minimized with different labels over the course of training. It is also problematic for methods enforcing consistency because such methods will use the invalid sample to enforce consistency with respect to two different classes. We note that invalid samples will occur for any method that uses GGA and most methods will incur some irreducible error from training on an ambiguous sample.

Here, we discuss how inter-class and intra-class GED relate to number of invalid and overlapping samples. Let $\mathcal{I}$ be the set of all invalid samples, $\mathcal{O}$ the set of overlapping samples, and $\tilde{\mathcal{O}} := \mathcal{O} \setminus \mathcal{I}$ be the set of intra-class (valid) overlapping samples. Let $C'$ be the lower bound on the number of

invalid samples we computed in Claim A.1. $C'$ is controlled by two parameters, $\mathbf{D}$ and $\Gamma$. We see that whether samples are, on average, invalid or merely overlapping is dependent on the average distance between samples of different classes when $\Gamma$ is held constant. Clearly, when training, we desire that $\frac{|\mathcal{I}|}{|\tilde{\mathcal{O}}|} \to 0$, as this ensures the model mostly sees valid samples. We note that this ratio is proportional to inter-class and intra-class distances as follows:

Recall that if $\mathcal{A}(\overline{x}_i) \in \mathcal{I}$, $(\mathbf{D}_{ij} - \Gamma_i - \Gamma_j)|\mathbf{y}_i - \mathbf{y}_j| < 0$ or equivalently, $\mathbf{D}_{ij} < \Gamma_i + \Gamma_j$, for $\mathbf{y}_i \neq \mathbf{y}_j$. Now, if $\mathcal{A}(\overline{x}_i) \in \tilde{\mathcal{O}}$, $\mathbf{D}_{ij} < \Gamma_i + \Gamma_j$, for $\mathbf{y}_i = \mathbf{y}_j$. Then, $|\mathcal{I}| \sim \mathbb{1}(\mathbf{D}_{ij} < \Gamma_i + \Gamma_j)$, for $\mathbf{y}_i \neq \mathbf{y}_j$ and $|\tilde{\mathcal{O}}| \sim \mathbb{1}(\mathbf{D}_{ij} < \Gamma_i + \Gamma_j)$, for $\mathbf{y}_i = \mathbf{y}_j$.

Now, $\frac{|\mathcal{I}|}{|\tilde{\mathcal{O}}|} \sim= \frac{\mathbb{1}(\mathbf{D}_{ij} < \Gamma_i + \Gamma_j), \text{ for } \mathbf{y}_i \neq \mathbf{y}_j}{\mathbb{1}(\mathbf{D}_{ij} < \Gamma_i + \Gamma_j), \text{ for } \mathbf{y}_i = \mathbf{y}_j} \to 0$, when inter-class distance is large for many samples, (i.e. the numerator is minimized), and when the intra-class distance is small for many samples (the denominator is maximized). This suggests it is desirable to have a lower average intra-class distance and a higher average inter-class distance.

While GED between samples cannot be controlled, the augmentation strength, $\Gamma$, can be controlled. It is desirable to minimize the number of invalid samples, while simultaneously maximizing the number of valid (including overlapping) augmented samples as follows:

$$\min_{\Gamma}(C) \text{ s.t. } \max_{\Gamma}\left(\sum_{\overline{x} \in \mathcal{X}} |\mathcal{B}(\overline{x})|\right)$$

While the above optimization is intractable and assumes label information, it alludes to two properties critical to the success of contrastive learning: connectedness of samples and recoverability (HaoChen et al., 2021). The number of invalid samples is indicative of the recoverability of different classes, while the above optimization indicates that we must also consider how well connected the augmentation sets are. We formalize this discussion in the next section.

### A.1.3 GGA AND THE POPULATION AUGMENTATION GRAPH

The preceding section discusses the relationship between GGA, GED and error introduced by invalid samples. However, this analysis is method-agnostic and does not offer theoretical insights into graph contrastive learning.

In computer vision, recent attempts to analyze theoretically the performance of contrastive learning often assumes that sample views are independent, a condition clearly violated by data augmentation (Arora et al., 2019; Tosh et al., 2021). To avoid this assumption, HaoChen et al. (2021) recently introduced the notion of a *population augmentation graph* (PAG), which represents augmented samples as nodes and weighted edges as the likelihood of generating a given pair of augmented samples from the same clean sample. Because samples from the same class are more likely to produce the same augmented sample than two random classes, connected subgraphs or communities in the PAG naturally correspond to underlying classes. HaoChen et al. (2021) designed and theoretically analyzed a CL objective that performed spectral decomposition on the PAG to recover these subgraphs (classes). Using their proposed objective and the PAG, they were able to provide the first accuracy guarantees for CL.

We begin by defining the PAG and the assumptions critical to HaoChen et al. (2021)'s analysis. Then, we extend our analysis from the preceding section to discuss how well these assumptions are supported for GCL.

**Definition 4** (Population Augmentation Graph (HaoChen et al., 2021))**.** *Given a natural dataset $\overline{\mathcal{X}}$, let $\mathcal{A}(\cdot|\overline{x})$ be the distribution of augmentations given a natural sample $\overline{x}$, or, intuitively, as the probability of generating a particular augmented sample from the large but finite set of all possible augmented versions of $\overline{x}$. Then, $\mathcal{X} := \cup_{\overline{x} \in \overline{\mathcal{X}}} \mathcal{A}(\cdot|\overline{x})$.*

*Let $\mathcal{G}^p$ be the population augmentation graph, where all $N$ samples in $\mathcal{X}$ form the nodes and $W \in \mathbb{R}^{N \times N}$ is the corresponding adjacency matrix. The edge weight between two nodes $x$ and $x'$ is defined as*

$$w_{x,x'} := \mathbb{E}_{\overline{x} \in \mathcal{P}_{\overline{\mathcal{X}}}}[\mathcal{A}(x|\overline{x})\mathcal{A}(x'|\overline{x})].$$

*Intuitively, if $w_{x,x'}$ is larger, it is relatively easier to generate the augmented pair from the same natural sample.*

Now, since graphs are discrete, the augmentation severity is restricted and only one edit can be applied at a time, we can completely define the population augmentation graph. Specifically, by using Remark A.1, the entire set of allowable augmentations can be determined. Moreover, recall that augmentations are performed randomly. Therefore, any $x \in \mathcal{B}(\overline{x}_i)$ is equally likely, so $\mathcal{A}(x|\overline{x}_i) = \frac{1}{|\mathcal{B}(x)_i|}$. However, if $x' \not\ni \mathcal{B}(\overline{x}_i)$, $\mathcal{A}(x|\overline{x}_i) = 0$ because it is not considered an allowable augmentation for $\overline{x}_i$. Note $w_{x,x'} > \frac{1}{|\mathcal{X}|} \frac{1}{|\mathcal{B}(\overline{x}_i)|^2}$ when $x, x'$ are both *overlapping samples*, i.e. $x \in \mathcal{B}(\overline{x}_j), x' \in \mathcal{B}(\overline{x}_j)$ for $i \neq j$. We refer to an edge whose endpoints are both overlapping samples as an *overlapping edge*. Similarly, a node in the PAG that is an overlapping sample is referred to as an *overlapping node*. As such, we have defined all possible nodes in $\mathcal{G}^p$ as well as how the edges are defined.

### A.1.4 Exploring PAG Structure

**Claim A.2.** *(Node Degree) Let $x$ be an overlapping node in the PAG. Additionally, suppose there is an alternative PAG, where $\tilde{x}$ is no longer an overlapping node but otherwise the PAG is the same. Then, $x$ will have a larger degree than $\tilde{x}$. This is true even if $\tilde{x}$ is not in an overlapping edge.*

*Proof.* Because $x$ is an overlapping node, $x \in \mathcal{B}(\overline{x}_i) \wedge x \in \mathcal{B}(\overline{x}_j)$ for some $i \neq j$. Then, $w_x = \sum_{x'} w_{xx'} = \sum_{x' \in \mathcal{B}(\overline{x}_i)} w_{xx'} + \sum_{x' \in \mathcal{B}(\overline{x}_j)} w_{xx'}$. Now, in the alternative PAG, $\tilde{x}$ is not an overlapping node, so $\tilde{x} \in \mathcal{B}(\overline{x}_i) \wedge \tilde{x} \notin \mathcal{B}(\overline{x}_j), \forall j \neq i$. Then $w_{\tilde{x}} = \sum_{x'} w_{\tilde{x}x'} = \sum_{x' \in \mathcal{B}(\overline{x}_i)} w_{xx'}$. Clearly, $w_x > w_{\tilde{x}}$. This that if a sample is an overlapping node, it will have a higher degree than if the same sample were not an overlapping node. $\square$

**Claim A.3** (GED Influences PAG structure). *If data points $x, x'$ share an edge in the PAG, then $\max\left(GED(x, \overline{x}_i), GED(x', \overline{x}_i)\right) < \max\{\gamma|\mathcal{V}_i|, \gamma|\mathcal{E}_i|\}$.*

*Proof.* $w_{xx'} > 0$ if and only if $x \in \mathcal{B}(\overline{x}_i) \wedge x' \in \mathcal{B}(\overline{x}_i)$. Recall in Def. 1, that $x \in \mathcal{B}(\overline{x}_i)$ if and only if $GED(x, \overline{x}_i) < \max\{\gamma|\mathcal{V}_i|, \gamma|\mathcal{E}_i|\}$, and similarly for $x'$. $\square$

Moreover, edge weights and node degrees are also influenced by the GED between samples. Overlapping edges can increase the weight between nodes. However, as discussed above, this requires that both ends of the edge are overlapping nodes. In Definition 2 and Remark 3, we show how GED can be used to determine the existence of such nodes. This further demonstrates the structure of the PAG is directly influenced by the GED between samples in $\mathcal{X}$.

We emphasize that our analysis suggests that practitioners may be using generic graph augmentations without realizing that they are implicitly assuming that GED is a useful metric for their problem.

Having elucidated the structure of the PAG and its relationships to GED, we discuss how its structure relates to the assumptions made by HaoChen et al. (2021) when analyzing the PAG. Namely, they require that the PAG "cannot be partitioned into too many disconnected sub-graphs", and that "labels are recoverable from augmentations." Indeed, their resulting bound on the error of spectral contrastive learning on the PAG depends upon the sparsest $m$-partition and classifier error.

The following assumption is from HaoChen et al. (2021):

**Assumption 1.** *(Labels are recoverable from augmentations). Let $\overline{x} \sim \mathcal{P}_{\overline{\mathcal{X}}}$ and $\mathbf{y}_{\overline{x}}$ be its label. Let the augmentation $x \sim \mathcal{A}(\cdot \mid \overline{x})$. We assume that there exists a classifier $g$ that can predict $\mathbf{y}_{\overline{x}}$ given $x$ with error at most $\alpha$. That is, $g(x) = \mathbf{y}_{\overline{x}}$ with probability at least $1 - \alpha$.*

**Claim A.4.** *(Recoverability is lower-bounded by the number of invalid samples). $\alpha$ (Assumption 1) can be lower-bounded when when $\mathcal{B}(\overline{x})$ contains an invalid sample: $\alpha \geq \frac{1}{\mathcal{B}(\overline{x})} - \frac{1}{|\mathcal{B}(x)|\tilde{Y}|}$, where where $\tilde{Y}$ is the set of labels represented among the natural samples that may have generated $x$.*

*Proof.* For this claim, we first discuss the best error that can be expected when classifying an invalid sample, and then we discuss the likelihood of encountering such a sample given some $\overline{x}$. Let $x$ be an

invalid sample that can be generated from natural samples: $\tilde{X} = \{(\overline{x}_1, \mathbf{y}_{\overline{x}_1}), \ldots (\overline{x}_k, \mathbf{y}_{\overline{x}_k})\}$. Clearly, $x$'s label is not well defined as it could be assigned any label $\tilde{y} \in \tilde{Y}$. However, the classifier $g$, is assumed to predict $g(x) = \mathbf{y}_{\overline{x}_i}, \forall \overline{x}_i \in \tilde{X}$ with error at most $\alpha$. Then, the minimum error for such a classifier is $1 - \frac{1}{|\tilde{Y}|}$, since the classifier does not know which natural sample generated $x$. For the remainder of the proof, we assume that $g$ can correctly classify all samples expect invalid samples to derive a lower bound.

Having established the minimum error of a classifier on an invalid sample, we determine how likely $g$ is to encounter such a sample given $\overline{x}$. Note that by assuming that $g$ can correctly identify all other augmented samples, the classifier error is only incurred when $x$ is an invalid sample. Through Remark 3, we first determine if an invalid sample is possible given a particular $\overline{x}$. If an invalid sample is possible, recall that every sample in the augmentation set, $\mathcal{B}(\overline{x})$, is equally likely by definition of generic graph augmentations. Therefore, the likelihood of generating $x$ given $\overline{x}$ is $\sim \frac{1}{|\mathcal{B}(\overline{x})|}$, where the size of the augmentation set can be determined using Remark A.1. We note that we could not provide an exact likelihood here because we assume (i) isomorphic graphs are counted separately and (ii) there is only one invalid sample in $\mathcal{B}(x)$, when in practice there may be multiple invalid samples. Nonetheless, we are able to derive a lower-bound on the error of the classifier, $g$, given a particular $\overline{x}$, by considering the likelihood of encountering an invalid sample and the error such a sample incurs: $\frac{1}{\mathcal{B}(x)} - \frac{1}{|\mathcal{B}(x)||\tilde{Y}|} \leq \alpha$. While the above analysis focus on a particular $\overline{x}$, we can extend the analysis to consider all samples, if we establish the likelihood of selecting a natural sample that can produce an invalid sample. (See subsubsection A.1.2 for related discussion.) □

Lastly, we hypothesize that $GED$ can be related to the Dirichlet conductance of the PAG, where Dirichlet conductance measures how many edges cross between a subset, $S$, and its complement relative to the total number of edges in the subset. We discuss our intuition in the following simple example, but leave a rigorous mathematical discussion to future work. Let $\overline{\mathcal{X}}$ be a dataset such that $\min_{i \neq j} GED(x_i, x_j) > \max(\Gamma)$, i.e the minimum distance between any two samples in the dataset is greater than the maximum allowable edits. Then, clearly the PAG contains $|\overline{\mathcal{X}}|$ fully connected subgraphs (cliques) that correspond to $\mathcal{B}(\overline{x})$, where $w_{xx'} = \frac{1}{|\mathcal{B}(\overline{x})|}$ for $x, x' \in \mathcal{B}(\overline{x})$. Given the structure of the graph, the conductance is minimized when $S = \mathcal{B}(\overline{x})$, as all edges within the subset are already contained. There are no edges to the complement because there are no overlapping samples by construction. We suspect that this observation can be extended to understand the behavior of the sparsest m-partition of the PAG, which HaoChen et al. (2021) use in their error bounds, but we leave that analysis to future work.

## A.2 DATASET GENERATION AND EXPERIMENTAL DETAILS

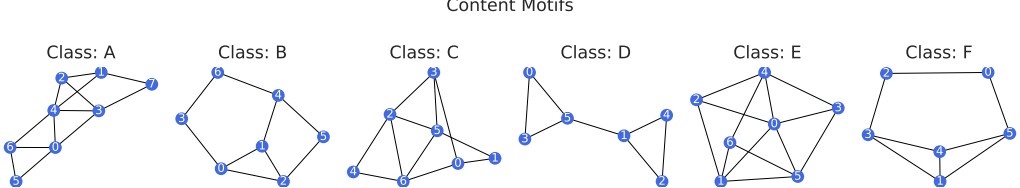

Figure 5: **Motifs used to determine class labels.**

We use the motifs shown in Fig. A.2 to define a 6 class graph classification task. It is important to ensure that the motifs are not isomorphic, as many GNNs are less expressive than the 1-Weisfeiler Lehman's test for isomorphism (Xu et al. (2019)). For each class, 1000 random samples are generated as follows: (i) We randomly select between 1-3 motifs to be in each sample. At this time, motifs all belong to the same class, though this condition could easily be changed for a more difficult task. (ii) We define the number of content nodes, $C_n$, as the size of the selected motif, scaled by the number of motifs in the sample. (iii) For a given style ratio, we determine the number of possible style nodes as $S_n = \rho C_n$ (iv). We define $RBG(n)$ using networkx's [1] random tree generator:

---
[1] https://networkx.org/documentation/stable/

`networkx.generators.trees.random_tree`. We note that other random graph genera-tors would also be well suited for this task. (v) For additional randomness, we create background graphs using $S_n \pm 2$, and also randomly perturb up-to 10% of edges in sample. We repeat this set-up with $\rho \in \{0.5, 1.0, 1.5, 2.0, 2.5, 3.0\}$ to generate the datasets used in Sec 5.

*Experimental Set-up:* We follow You et al. (2020a) for TUDataset experiments. We use a 5-layer GIN model with sum pooling for all synthetic experiments. Models are pretrained for 100 epochs and then fine-tuned for 200 epochs with 1 learning rate drop when the loss plateaus. The hidden layer dimension is 32. We concatenate hidden representations for a representation dimension of 160. All models are trained with Adam, lr = 0.001. For the sample complexity experiment, we allow for end to end training. For all other experiments, we freeze the backbone and only train the linear prediction head.

## A.3 INDUCTIVE BIAS, ADDITIONAL RESULTS

To further demonstrate the effectiveness of untrained models on popular benchmarks, we include results from different GNN architectures: GraphSage (Hamilton et al. (2017)), PNA (Corso et al. (2020)), GCN (Kipf & Welling (2017)) and GAT (Velickovic et al. (2018)). We note that while it has been informally discussed that untrained GNNs have a strong inductive bias, our intention is to formalize why these untrained models *must* be included when evaluating unsupervised graph representation learning. Moreover, in Tab. A.4, we include a variant of untrained models, where we initialize BatchNorm statistics without computing any gradients by iterating over the dataset once. For several datasets and baselines, we see that this Warmup step makes the untrained baseline even stronger. We believe that future work should also consider this simple baseline when evaluating the performance of their models.

Table 3: Inductive Bias.

| GraphSAGE | 3 Layer | 4 Layer | 5 Layer | GraphCL | InfoGraph |
|---|---|---|---|---|---|
| MUTAG | $0.85 \pm 0.005$ | $0.85 \pm 0.006$ | $0.85 \pm 0.005$ | $0.82 \pm 0.040$ | $0.85 \pm 0.005$ |
| PROTEINS | $0.73 \pm 0.004$ | $0.73 \pm 0.003$ | $0.74 \pm 0.005$ | $0.75 \pm 0.002$ | $0.74 \pm 0.008$ |
| NCI1 | $0.74 \pm 0.003$ | $0.75 \pm 0.006$ | $0.73 \pm 0.011$ | $0.78 \pm 0.000$ | $0.79 \pm 0.002$ |
| DD | $0.77 \pm 0.006$ | $0.78 \pm 0.002$ | $0.78 \pm 0.005$ | $0.80 \pm 0.008$ | $0.77 \pm 0.010$ |
| REDDIT-B | $0.85 \pm 0.014$ | $0.83 \pm 0.016$ | $0.83 \pm 0.005$ | – | $0.66 \pm 0.137$ |
| IMDB-B | $0.66 \pm 0.012$ | $0.81 \pm 0.008$ | $0.81 \pm 0.008$ | – | – |

| PNA | 3 Layer | 4 Layer | 5 Layer | GraphCL | InfoGraph |
|---|---|---|---|---|---|
| MUTAG | $0.88 \pm 0.011$ | $0.88 \pm 0.010$ | $0.89 \pm 0.009$ | $0.86 \pm 0.023$ | $0.90 \pm 0.014$ |
| PROTEINS | $0.74 \pm 0.003$ | $0.74 \pm 0.012$ | $0.74 \pm 0.005$ | $0.74 \pm 0.007$ | $0.74 \pm 0.003$ |
| NCI1 | $0.67 \pm 0.008$ | $0.68 \pm 0.011$ | $0.68 \pm 0.010$ | $0.78 \pm 0.008$ | $0.77 \pm 0.019$ |
| DD | $0.76 \pm 0.014$ | $0.76 \pm 0.002$ | $0.76 \pm 0.008$ | $0.80 \pm 0.008$ | $0.76 \pm 0.006$ |
| REDDIT-B | $0.90 \pm 0.003$ | $0.88 \pm 0.014$ | $0.89 \pm 0.010$ | $0.92 \pm 0.006$ | $0.92 \pm 0.006$ |
| IMDB-B | $0.72 \pm 0.007$ | $0.68 \pm 0.011$ | $0.68 \pm 0.010$ | $0.71 \pm 0.009$ | $0.71 \pm 0.009$ |

| GCN | 3 Layer | 4 Layer | 5 Layer | GraphCL | InfoGraph |
|---|---|---|---|---|---|
| MUTAG | $0.85 \pm 0.003$ | $0.85 \pm 0.004$ | $0.85 \pm 0.005$ | $0.82 \pm 0.013$ | $0.85 \pm 0.003$ |
| PROTEINS | $0.74 \pm 0.003$ | $0.73 \pm 0.007$ | $0.74 \pm 0.004$ | $0.75 \pm 0.004$ | $0.75 \pm 0.003$ |
| NCI1 | $0.76 \pm 0.004$ | $0.75 \pm 0.001$ | $0.75 \pm 0.002$ | $0.78 \pm 0.008$ | $0.79 \pm 0.007$ |
| DD | $0.78 \pm 0.002$ | $0.77 \pm 0.012$ | $0.78 \pm 0.003$ | $0.79 \pm 0.007$ | $0.76 \pm 0.003$ |
| REDDIT-B | $0.52 \pm 0.005$ | $0.51 \pm 0.003$ | $0.52 \pm 0.005$ | $0.92 \pm 0.002$ | $0.80 \pm 0.062$ |
| IMDB-B | $0.54 \pm 0.001$ | $0.57 \pm 0.016$ | $0.58 \pm 0.008$ | $0.71 \pm 0.011$ | $0.62 \pm 0.070$ |

| GAT | 3 Layer | 4 Layer | 5 Layer | GraphCL | InfoGraph |
|---|---|---|---|---|---|
| MUTAG | $0.84 \pm 0.003$ | $0.85 \pm 0.009$ | $0.84 \pm 0.003$ | $0.81 \pm 0.032$ | $0.85 \pm 0.013$ |
| PROTEINS | $0.74 \pm 0.002$ | $0.74 \pm 0.005$ | $0.74 \pm 0.006$ | $0.74 \pm 0.007$ | $0.74 \pm 0.005$ |
| NCI1 | $0.76 \pm 0.009$ | $0.75 \pm 0.004$ | $0.76 \pm 0.002$ | $0.78 \pm 0.004$ | $0.70 \pm 0.040$ |
| DD | $0.78 \pm 0.005$ | $0.77 \pm 0.006$ | $0.79 \pm 0.001$ | $0.79 \pm 0.003$ | $0.76 \pm 0.005$ |
| REDDIT-B | $0.52 \pm 0.005$ | $0.53 \pm 0.004$ | $0.52 \pm 0.012$ | $0.75 \pm 0.004$ | – |
| IMDB-B | $0.51 \pm 0.004$ | $0.51 \pm 0.009$ | $0.50 \pm 0.005$ | $0.51 \pm 0.007$ | – |

## A.4 INVARIANCE, ADDITIONAL RESULTS

We extend our representation invariance results on standard benchmarks to different architectures below in Tab. A.4. As in our main results, we use random subgraph sampling and node drop-ping as our augmentations, following You et al. (2020a), when computing invariance. We find that

similar trends hold: while training with GCL does improve performance and invariance somewhat, untrained models perform comparably without the same levels of invariance.

Table 4: Invariance Table.

|  | RandGAT | (Acc) | WarmupGAT | (Acc) | GAT (GraphCL) | (Acc) |
|---|---|---|---|---|---|---|
| MUTAG | 0.993 | 0.843 | 0.364 | 0.793 | 0.608 | 0.807 |
| PROTEINS | 0.987 | 0.737 | 0.819 | 0.738 | 0.554 | 0.744 |
| NCI1 | 0.993 | 0.761 | 0.543 | 0.771 | 0.669 | 0.781 |
| DD | 0.970 | 0.779 | 0.381 | 0.778 | 0.361 | 0.793 |
| REDDIT-B | 1.000 | 0.517 | 0.850 | 0.724 | 0.982 | 0.747 |
| IMDB-B | 1.000 | 0.512 | 0.979 | 0.670 | 0.994 | 0.512 |
|  | RandGIN | (Acc) | WarmupGIN | (Acc) | GIN (GraphCL) | (Acc) |
| MUTAG | 0.921 | 0.867 | 0.208 | 0.866 | 0.852 | 0.868 |
| PROTEINS | 0.910 | 0.745 | 0.495 | 0.750 | 0.547 | 0.744 |
| NCI1 | 0.921 | 0.707 | 0.281 | 0.769 | 0.768 | 0.778 |
| DD | 0.907 | 0.732 | 0.071 | 0.760 | 0.638 | 0.786 |
| REDDIT-B | 0.906 | 0.723 | 0.242 | 0.768 | 0.286 | 0.895 |
| IMDB-B | 0.914 | 0.672 | 0.791 | 0.700 | 0.468 | 0.711 |
|  | RandGCN | (Acc) | WarmupGCN |  | GCN (GraphCL) | (Acc) |
| MUTAG | 0.996 | 0.847 | 0.491 | 0.807 | 0.561 | 0.821 |
| PROTEINS | 0.980 | 0.739 | 0.886 | 0.750 | 0.765 | 0.749 |
| NCI1 | 0.991 | 0.756 | 0.480 | 0.767 | 0.664 | 0.780 |
| DD | 0.968 | 0.779 | 0.440 | 0.772 | 0.367 | 0.789 |
| REDDIT-B | 0.999 | 0.519 | 0.129 | 0.833 | 0.678 | 0.919 |
| IMDB-B | 0.914 | 0.540 | 0.539 | 0.833 | 0.994 | 0.709 |
|  | RandSAGE | (Acc) | WarmupSAGE | (Acc) | SAGE (GraphCL) | (Acc) |
| MUTAG | 0.910 | 0.846 | 0.273 | 0.801 | 0.303 | 0.823 |
| PROTEINS | 0.907 | 0.732 | 0.582 | 0.747 | 0.507 | 0.749 |
| NCI1 | 0.912 | 0.737 | 0.412 | 0.771 | 0.579 | 0.779 |
| DD | 0.590 | 0.771 | 0.590 | 0.781 | 0.727 | 0.801 |
| REDDIT-B | 0.833 | 0.849 | 0.225 | 0.740 | – | – |
| IMDB-B | 0.223 | 0.663 | 0.223 | 0.497 | – | – |

## A.5 DATASET STATISTICS

## A.6 RELATED WORK

*Graph Data Augmentation:* Augmentations for graphs are difficult to define due to their discrete, non-euclidean nature. Furthermore, unlike images or natural language where there is an intuitive understanding of what changes will preserve task-relevant information, no such understanding exists for graphs. Indeed, a single edge change can completely change the properties of a molecular graph. Therefore, only a few works consider graph data augmentation. Zhao et al. (2020) note that a node classification task can be perfectly solved if edges only exist between same class samples. They train a neural edge predictor to increase homophily by adding edges between nodes expected to be of the same class and break edges between nodes of expected dissimilar classes. However, this approach is expensive and not applicable to graph classification. Kong et al. (2020) argue that information preserving topological transformations are difficult for the aforementioned reasons and instead focus on feature augmentations. Throughout training, they add an adversarial perturbation to node features to improve generalization. To avoid incurring the large expense of adversarial training, they leverage Shafahi et al. (2019) and compute the gradient of the model weights while computing the gradients of the adversarial perturbation. This approach is not directly applicable to contrastive learning, where label information cannot be used to generate the adversarial perturbation.

*Graph Self-Supervised Learning:* Several paradigms for self-supervised learning in graphs have been recently explored, including the use of pre-text tasks, multi-tasks, and unsupervised learning. See Liu et al. (2021) for an up-to-date survey. Graph pre-text tasks are often reminiscent of image in-painting tasks Yu et al. (2018), and seek to complete masked graphs and/or node features (You et al. (2020b); Hu et al. (2020)). Other successful approaches include predicting graph level or property level properties during pre-training or part of regular training to prevent over-fitting (Hu et al. (2020)). These tasks often must be carefully selected to avoid negative transfer between tasks. Many

Table 5: *Dataset Description*

| Name | Graphs | Classes | Avg. Nodes | Avg. Edges | Domain |
|---|---|---|---|---|---|
| IMDB-BINARY  (Yanardag & Vishwanathan, 2015) | 1000 | 2 | 19.77 | 96.53 | Social |
| REDDIT-BINARY  (Yanardag & Vishwanathan, 2015) | 2000 | 2 | 429.63 | 497.75 | Social |
| MUTAG  (Kriege & Mutzel, 2012) | 188 | 2 | 17.93 | 19.79 | Molecule |
| PROTEINS  (Borgwardt et al., 2005) | 1113 | 2 | 39.06 | 72.82 | Bioinf. |
| DD  (Shervashidze et al., 2011) | 1178 | 2 | 284.32 | 715.66 | Bioinf. |
| NCI1  (Wale & Karypis, 2006) | 4110 | 2 | 29.87 | 32.30 | Molecule |

Table 6: **Selected Graph Contrastive Learning Frameworks.** Brief description of augmentations used by selected frameworks is provided. Most frameworks use random corruptive, sampling, or diffusion-based approaches to generate augmentations.

| Method | Augmentations |
|---|---|
| GraphCL (You et al. (2020a)) | Node Dropping, Edge Adding/Dropping, Attr. Masking, Subgraph Extraction |
| GCC (Qiu et al. (2020)) | RWR Subgraph Extraction of Ego Network |
| MVGRL (Hassani & Ahmadi (2020)) | PPR Diffusion + Sampling |
| GCA (Zhu et al. (2020)) | Edge Dropping, Attr. Masking (both weighted by centrality) |
| BGRL (Thakoor et al. (2021)) | Edge Dropping, Attr. Masking |
| SelfGNN (Kefato & Girdzijauskas (2021)) | Attr. Splitting, Attr. Standardization + Scaling, Local Degree Profile, Paste + Local Degree Profile |

unsupervised approaches have also been proposed. Sun et al. (2020); Velickovic et al. (2019) draw inspiration from Hjelm et al. (2019) and maximize the mutual information between global and local representations. MVGRL (Hassani & Ahmadi (2020)) contrasts different views at multiple granularities similar to van den Oord et al. (2018). You et al. (2020a); Qiu et al. (2020); Zhu et al. (2020); Thakoor et al. (2021); Kefato & Girdzijauskas (2021) use augmentations to generate views for contrastive learning. See Table A.6 for a summary of the augmentations used. We note that random corruption, sampling or diffusion based approaches often do not preserve task relevant information or introduce meaningful invariances.

