# OpenReview forum: "Interrogating Paradigms in Self-supervised Graph Representation Learning"
_ICLR.cc/2022/Conference — ICLR 2022 Submitted_

### Official Review · Reviewer_MoyX · 2021-11-01

**Correctness:** 2
**Technical Novelty And Significance:** 2
**Empirical Novelty And Significance:** 2
**Recommendation:** 5
**Confidence:** 4

**Main Review:**

One of the claimed contributions is the “theoretical analysis of when contrastive learning is expected to work well”. However, in section 3.2, there is no link of the “fixed error” of the contrastive learning performance to the graph properties.

A conclusion is made from Fig 1 (a), “the learning invariance was not necessary for downstream task performance”.  However, there does exist performance (accuracy) improvement on dataset of IMDB-BINARY and NCI1.  It would be more valuable to study the necessarity of learning invariance on different types of graphs, and how this necessarity depends on what kinds of graph properties.

Another difficulty is found in understanding the claimed contribution to “identify a style vs. content trade-off in graphs and introduce an extensive benchmark setup that can carefully control this trade-off”.  Section 5 did analysis of style (irrelevant information) vs. content (task relevant information). However, it is still unclear about how to identify a trade-off parameter in a given application problem.

More details should be given about datasets PROTEINS in evaluation, same for  MUTAG, IMDB-BINARY and IMDB-BINARY.



**Summary Of The Paper:**

This paper targets on investigating the role of dataset properties and augmentation strategies on the success of graph contrastive learning and reconstruction-based approaches.

**Summary Of The Review:**

The paper works on an interesting and valuable problem. However, the analysis results are not strong enough to support the claimed contributions.

---

> ### Author Response · Authors · 2021-11-16
> **Response to Reviewer MoyX**
>
> ***Thank you for thoughtful comments and finding the paper to work on an interesting and valuable problem. We address your concerns below:***
>
> + ***Concerns regarding analysis of GGA and GCL.***
>
> We have incorporated reviewer feedback to make this section more thorough and understandable. Please see the official comment on “Restructured Analysis” for a summary of our changes, and Appendix A in the main paper the improved version.
>
> Specifically, we restructured our discussion of fixed error as follows: we begin by defining invalid samples (augmented samples generated from natural samples of different labels), then we show how the existence of these invalid samples is determined by the GED between natural samples, and finally we demonstrate how one could estimate the number of invalid samples. We note that any representation learning method based on GGA will encounter invalid samples since they arise from dataset characteristics and the augmentation itself. We then define the population augmentation graph and discuss how GGA and GED relate to the “recoverability” assumption made by HaoChen et. al when analyzing CL. HaoChen et al. assume that there is a classifier that can recover the true label of an augmented sample with low error, and use this error in their subsequent error bounds for CL. If we assume that invalid samples are labeled incorrectly (it is unclear which natural sample’s label should be adopted), we are able to lower-bound the error of such a classifier by determining how likely it is that the classifier encounters an invalid sample.
>
> + ***Concerns regarding the necessity of invariance.***
>
> We agree that studying the necessity of learning invariance on different types of graphs in relation to the underlying graph properties is an interesting and promising direction for future work. Within the scope of our work, we believe our experiments on invariance are important because they demonstrate that training with GGA leads to improved invariance but does not greatly improve accuracy. A central tenet of visual CL is that augmentations (cropping, color jittering, rotation) introduce invariance to properties that are not relevant for the downstream task. From our empirical study with standard benchmarks, *we find that GGA does not always introduce such useful invariances*. Furthermore, in Section 5, we create oracle, context-aware augmentations that are designed to directly introduce invariance to the background or “style” graph. Here, as the style ratio increases, it is important to learn invariance to the background graph so GNNs can correctly identify the content graphs. *While this clearly evidences the benefits of inducing the most appropriate invariances in the representations, the persisting gap between the performance of vanilla supervised learning and graph URL warrants further analysis.*
>
> + ***Concerns regarding style vs. content.***
>
> While the proposed synthetic data generation process enabled us to directly control the style vs. content ratio through the style-ratio parameter, we agree with the reviewer that it is difficult to directly identify such a parameter in a given application. However, such a partition does exist in natural datasets and is important to learning high-quality representations. For example, in molecular datasets, carbon rings can be seen as style and functional groups can be seen as content as they are important when predicting molecular properties. When using appropriate data augmentations, it is possible to learn representations that preserve content, while becoming invariant to style, even if the trade-off parameter is not exactly known. Domain knowledge can be used to design such augmentations. For example, augmentations that drop carbon rings from molecular graphs but do not modify functional groups would respect this partition. However, in Section 5, we show that even with oracle, context-aware augmentation, the performance of unsupervised methods lags behind that of supervised learning. This suggests that other components of the unsupervised graph representation learning pipeline must also be improved if we are to match/surpass supervised performance.
>
> + ***Concerns about dataset information.***
>
> We have added an additional table (Table 5.) to the appendix that contains dataset information.

---

> > ### Comment · Reviewer_MoyX · 2021-11-29
> > **Thanks for the reply.**
> >
> > Thanks for the reply. I basically like this work. However, I decided not to change the score, because I  think the paper will be much more valuable if it studies also the necessarity of learning invariance on different types of graphs, and how this necessarity depends on what kinds of graph properties.

---

### Official Review · Reviewer_UyEA · 2021-11-02

**Correctness:** 1
**Technical Novelty And Significance:** 2
**Empirical Novelty And Significance:** 2
**Recommendation:** 5
**Confidence:** 4

**Main Review:**

Recently, we have witnessed an explosive development of self-supervised graph learning methods. As one of the most import parts, data augmentation technology provides a flexible way to generate training set without the high-cost labels. Many works draw lessons from the computer vision community and design a bunch of methods to generate the augmented views for contrastive learning graph representations. However, it's still not clear why and to what extent the data-augmentation methods have positive benefits for directing the graph neural models to the right way. There're some works trying to uncover the mystery by studying the contrastive loss, but rare attention being paid to the impact of the graph augmentation. While this work seems to be a pilot study about the condition when contrastive graph learning can work well. From the manuscript, we can see that authors present the relation between graph edit distance and augmentation operations (e.g. dropping edges/nodes, adding edges/nodes), and claim that the performance of contrastive graph learning could be bounded by the graph edit distance (GED) between different classes.

Concerns:
1. The studied problem in this work is very important for the research community. However, the quality of the presentation make it difficult to understand the motivation behind. After checking the draft several times, it's still unclear why the GED can be a general criterion to judge the success of GraphCL with both node attributes and graph structure as the input data. As we know that GED is defined on the graph structure. I just wonder the generalizability of the claim raised by this work. Suppose that the GED has positive impact, the writing is difficult to follow. There're many claims which are copied from the previous work proposed by HaoChen et al.. While they are not organized logically, and the missing background knowledge makes some claims difficult to capture the logic behind. For example, what is the definition of spectral decomposition, which is presented in HaoChen et al. But it shows up in Section 3 without any explanation.

2. Besides the discussion about the connection GED and GraphCL, the contribution of this work seems to be applied the augmentation method to reconstruction-based task like edge reconstruction. However, it seems not to be discussed in the main content until the section for presenting the experiment setting.



**Summary Of The Paper:**

This paper attempts to study the connection between graph edit distance and the reason behind the success of self-supervised graph learning.

**Summary Of The Review:**

Good research problem, but the presentation is not good enough to make the key idea clear.

---

> ### Author Response · Authors · 2021-11-16
> **Response to Reviewer UyEA**
>
> ***Thank you for your helpful feedback and finding the studied problem very important to the research community. We address your concerns below:***
>
> + ***Concerns regarding analysis of GED, GGA and PAG.***
>
> We have incorporated reviewer feedback to make this section more thorough and understandable. Please see the official comment on “Restructured Analysis” for a summary of our changes, and Appendix A in the main paper for the improved version.
>
> Specifically, we have reorganized our discussion of GED + GGA and GED + GGA + PAG into two parts, better contextualized relevant claims from HaoChen et al and more rigorous mathematical discussion.  Thank you for the feedback!
>
> + ***Concerns on generality of GED.***
> Regarding the generality of GED when considering graphs with node attributes, we note that our formulation can be easily extended to graphs with categorical node attributes if we introduce an additional graph edit operator that performs a “replacement” whenever node attributes between graphs disagree. Then, the GED is the cost of structural changes and the number of disagreements between graph attributes. We have added this discussion to the updated Appendix A.1 as well. Categorical variables are common in molecular classification tasks, where attributes correspond to elements, and discrete node attributes are often used when analyzing GNNs [1].
>
> Lastly, we clarify that GED amongst intra-class and inter-class samples helps us understand the behavior of GGA, and the structure of the PAG, which in tur, helps us understand the performance of contrastive learning. We note if the GED is on average lower for intra-class samples than inter-class samples, GGA will produce fewer invalid samples which can indeed be seen as a “positive impact.” Please see Appendix A.1.2 for a related discussion. *Overall though, our analysis demonstrates that the central assumption of GGA, that small changes to a graph will not alter its semantics, is incomplete. We must also consider the GED when determining an augmentation strength that preserves semantics to understand the impact of GGA.*
>
> + ***Concerns regarding contributions.***
>
> Thank you for this feedback! We have incorporated reviewer feedback to make our contributions more clear and understandable. Please see the official comment on “Novelty/Significance” for an overview of our changes, and Section 3 in the main paper for the improved version.
>
> We emphasize that while we are the first to implement and benchmark augmentation augmented graph autoencoders are a strong reconstruction-based baseline, it is only one of our contributions. *Our main contributions center around our analysis, empirical and analytical, of different learning paradigms and augmentations. Specifically, we consider three levels of problem knowledge when studying different paradigms and augmentations*: no knowledge, some knowledge and complete knowledge. Our empirical study, which extensively benchmarks untrained GNNs across several different datasets, studies how powerful GNN inductive bias is when we cannot rely upon problem knowledge. We then assume we have some knowledge through the soft inductive biases encoded into generic graph augmentations. Our analysis in the updated A.1 uses GED to demonstrate when GGA could be problematic. Lastly, in Section 5, we assume full knowledge of our problem and use oracle, context aware augmentations on our novel, synthetic benchmark dataset. Even in this idealized setting, we see that the performance of CL and reconstruction-based approaches lags far behind that of supervised learning; this suggests other components of the unsupervised graph representation learning pipeline must be improved before supervised performance is surpassed.

---

> > ### Comment · Reviewer_UyEA · 2021-11-30
> > **Thanks for the reply.**
> >
> > I really appreciate the detailed feedback from authors. I basically like the focus of this work on a very important and challenging problem which can potentially help to better understand the limits of self-supervised learning under different conditions. I tend to update my points considering that parts of my concerns have been addressed well. But, I think that current version of manuscript still needs a major improvement to make it concise and easy to follow.

---

### Official Review · Reviewer_97Kb · 2021-11-02

**Correctness:** 2
**Technical Novelty And Significance:** 2
**Empirical Novelty And Significance:** 2
**Recommendation:** 5
**Confidence:** 4

**Main Review:**

This paper studies the success conditions of graph contractive learning (GCL) with generic graph augmentation (GGA) and claims the success depends on the graph edit distance between classes. It empirically show that reconstruction-based approaches (GAE and AAGAE) perform well in low graph style regimes and GCL with GGA benefits from moderate style.

The problems investigated in the paper is interesting and is significant to the unsupervised graph representation learning community. However, their claims lack rigorous mathematical and empirical demonstrations. In addition, the analysis is restricted to spectral contractive loss functions, and it is not clear how to extend the analysis to general GCL approaches. Further, the experiments were conducted on relatively limited methods, and it is not clear whether the results are applicable to more general approaches.


## Strength


1.	The presentation is easy to follow.
2.	The problem of identifying success conditions of graph contractive learning investigated in the paper is interesting.
3.	The experimental results, i.e. the impacts of the ratio of style and content on the performance of GCL, GAE, and AAGAE, are interesting and may provide some insights to the unsupervised graph representation learning community.

## Weakness



1. The claims of the paper mainly build  upon the theoretical results of a recent paper [1], whose details are omitted in the paper. Would be great to carefully discuss the relationship to the results of [1]
2.  The analysis in Section 3 is restricted to PAG and the spectral contrastive loss and there are no illustrations to show how to extend the analysis to general GCL approaches.
3. Additionally, the claim, that the success of GCL with GGA depends on the graph edit distance between classes, lacks rigorous mathematical and empirical evidence.
4.  The experiments were conducted for only three methods (i.e., GCL, GAE, AAGAE) and the analysis of paper is restricted to GCL with GGA. There is no indication whether their results can be applied to general GCL approaches or other reconstruction-based approaches.


**Summary Of The Paper:**

The paper studies the success conditions for graph contractive learning (GCL) with generic graph augmentation (GGA) and asserts that these conditions depend on the graph edit distance of samples within and across classes. Using empirical evidence, the authors demonstrate that reconstruction-based approaches (GAE and AAGAE) perform well at low graph style regimes, while GCL with GGA benefits from moderate graph style regimes.

**Summary Of The Review:**

This paper studies an important open problem,  but is lack of rigorous mathematical demonstrations and has limited empirical studies.

---

> ### Author Response · Authors · 2021-11-16
> **Response to Reviewer 97Kb**
>
> ***Thank you for thoughtful comments and finding that the paper studies an important open problem. We address each of your concerns below:***
>
> + ***Concerns regarding PAG-based analysis (1,3).***
>
> We have incorporated reviewer feedback to make this section more thorough and understandable. Please see the official comment on “Restructured Analysis” for a summary of our changes, and Appendix A in the main paper for the improved version.
>
> More details regarding [1] and rigorous mathematical discussion have been added. Thank you for the feedback!
>
> + ***Concerns on generalizability of PAG and Spectral Contrastive Learning (2).***
>
> Thank you for this feedback. In the updated Appendix A.1, we show that any method that uses GGA will encounter invalid samples at a rate that is dependent on the GED between inter-class samples. This suggests all methods incur some irreducible error by virtue of their augmentation strategy.
>
> Our analysis uses the PAG and Spectral Contrastive Loss because this is the first contrastive loss that has *provable accuracy guarantees* under a linear probe and *does not assume that augmentations are conditionally independent* given the input. Other losses used by general GCL do not have such a guarantee. However, [1] does note that several popular losses can be seen as modifications of the spectral contrastive loss. For example, they show that the SimCLR loss, which is also used by GraphCL, can be recovered through some algebraic manipulation and dropping a term. They note that the spectral contrastive loss has similar performance but is not dependent on batch size.  Removing batch dependence is especially useful for graph representation learning where benchmark dataset are often quite small.
> We agree with the reviewer that extending our analysis to arbitrary loss functions is an interesting and promising future direction. Indeed, GraphCL claims “[it] can be rewritten as a general framework unifying a broad family of contrastive learning methods on graph-structured data” including DGI, InfoGraph, and GMI. Given that SimCLR’s loss function is already related to the spectral contrastive loss, it is plausible that other GCL frameworks may also be related, and is definitely part of our future work. Nonetheless, to the best of our knowledge, our work is the first to perform such an analysis on GCL.
>
> + ***Concerns on generalizability of Experimental Results (4).***
>
> We expect that our results will persist across different GCL or reconstruction-based methods. To keep our experiments tractable, we decided to conduct our experiments using representative, strong baselines for each paradigm: GraphCL for contrastive learning, and GAE/AAGAE for reconstruction based approaches, and a suite of GNN architectures.  *Nonetheless, a key finding of our work is that even with oracle context aware augmentations, these strong unsupervised baselines are unable to match supervised performance. We expect this to hold true for other methods.* Furthermore, in addition to the methods presented here, we did try using an augmentation-augmented variational graph autoencoder but found the training to be too unstable. This is in contrast to computer vision, where the more complex variational model outperforms the simple autoencoder. Our results suggest that more customized approaches, which are specific to graph data, for augmentation or model design may be needed when building unsupervised graph representation learning pipelines. In a potential final version of the paper, we will include one or two more methods for both CL and reconstruction-based approaches.

---

> > ### Comment · Reviewer_97Kb · 2021-11-30
> > **Thanks for the response**
> >
> > I appreciate the response from the authors! Part of my comments are addressed.  I tend to keep my rating unchanged.

---

### Official Review · Reviewer_UMWA · 2021-11-03

**Correctness:** 4
**Technical Novelty And Significance:** 2
**Empirical Novelty And Significance:** 2
**Recommendation:** 5
**Confidence:** 4

**Main Review:**

Strengths: The motivation of this paper is relatively new. Data augmentation in graph contrastive learning is an important part which is often overlooked by researchers. It is a successful transfer of work from CV and can show that the author has a deep understanding of the latest work on CV contrastive learning. The benchmark design based on motif and the design of "STYLE VS. CONTENT" on the graph are both highlights.

Weaknesses: The analysis of the relationship between graph contrastive learning effective conditions and data augmentation is not theoretical enough. Some sentences confused me when reading such as "To define a min-cut partition, the number of edges crossing a subset of nodes should be minimized while the number of edges within a subset should be high". It would be better if the theoretical analysis part was supplemented in more detail. The problem and process of analysis in Section 3.2 is too hand-waving.
Secondly, I think your experiment with GNN without training is controversial in Table 1. If you follow the training mode of graph contrastive learning (first defined in DGI), the untrained GNN is combined with a trained downstream linear layer. I think the existence of this linear layer is not enough to support your conclusion that untrained GNN is competitive. If you insist on proving this statement, please use the results of the clustering experiment which can be found in MVGRL.
Finally, I think the contribution of the paper is not enough. Although the analysis and motivation are new, they are still used from CV. The design of benchmark and "STYLE VS. CONTENT" is novel but simple. I think it is a relatively complete work to propose a better data augmentation or sampling method on this benchmark based on the riched theory.


**Summary Of The Paper:**

This paper focuses on the augmentation of graph contrastive learning. Authors make further explorations on graph contrastive learning based on the previous CV contrastive work (HaoChen et al. (2021) ) and hope to design a good benchmark by analyzing the relationship between GCL/auto-encoding methods and augmentations. In the analysis part, authors mainly use the perspective of population augmentation graph work to find connections with existing graph augmentations (mainly You et al., 2020a) and analyze the relationship between GED and task performance/downstream labels. In the benchmark part, authors proposes a benchmark based on the idea of "STYLE VS. CONTENT" which is inherited from CV contrastive work. The analysis of the experimental results is okay.


**Summary Of The Review:**

The motivation of the article is good, but I have some problems in terms of contribution and theoretical analysis. Experiment and analysis are okay.

---

> ### Author Response · Authors · 2021-11-16
> **Response to Reviewer UMWA (Part 1)**
>
> ***Thank you for thoughtful comments and finding the motivation problem interesting. We address your concerns below:***
>
> + ***Concerns regarding analysis of GED, GGA and PAG***
>
> We have incorporated reviewer feedback to make this section more thorough and understandable. Please see the official comment on “Restructured Analysis” for a summary of our changes, and Appendix A in the main paper the improved version.
>
> + ***Concerns on inductive bias experiment.***
>
> While we do agree that a fully unsupervised approach using clustering is an alternative, our current setting, which only trains a classifier, is nonetheless still a valid approach for the following reasons: (i) The evaluation protocol is standard across many GCL frameworks including GraphCL, InfoGraph, and MVGRL. (ii) The clustering experiments in MVGRL were only conducted on node classification datasets, not graph classification as we consider here. (iii) The intention of this experiment is to demonstrate the strong inductive bias of untrained GNNs. Given that the same type of classifier is used for all methods, we can still evaluate the inductive bias of the untrained model. While using k-means to perform clustering would evaluate the separability of the representations in Euclidean space, we evaluate the separability of the representations in a higher dimensional space by using the classifier.
>
> + ***Concerns on contribution and relationship to CV.***
>
> We have incorporated reviewer feedback and improved our paper to better represent our contributions. Please see our official comment on novelty and significance, as well as the updated paper, where we discuss the significance of our contribution. Below, we highlight how our contributions and motivations are different from that of CV.
>
> Many popular GCL frameworks are variants of successful frameworks in computer vision with modifications to make them suitable for graphs. For example, GraphCL resembles SimCLR, MVGRL resembles CMC, InfoGraph resembles InfoMax, and concurrent work, Bootstrapped Graph Latents closely resembles BYOL. Therefore, we argue that extending analysis from CV to understand GCL frameworks is important to understand if (i) the assumptions critical to CV analysis extend to GCL and (ii) how graph-specific modifications affect existing analysis.
>
> Indeed, You et al. proposed generic graph augmentations as a graph-specific analog to the aggressive data augmentation used by SimCLR. However, in Appendix A.1.4, we identify a simple situation where GGA can induce behavior that does not align with assumptions made in visual CL. Namely, we are able to lower-bound the error of classifiers on augmented samples, given graph edit distance between samples and label information. This suggests that the “recoverability” assumption need not hold for GCL with GGA.

---

> > ### Author Response · Authors · 2021-11-16
> > **Response to Reviewer UMWA (Part 2)**
> >
> > + ***Concerns on Style vs. Content Benchmark.***
> >
> > *The simplicity of the proposed benchmark is intentional and allows for flexible modifications to increase complexity.* We first reiterate the limitations addressed by our proposed benchmark and then discuss several extensions to our data generation process that can increase complexity.
> > While more complex datasets can be constructed, our proposed benchmark nonetheless *addresses several limitations of existing datasets* that enable careful evaluation of different methods and augmentations. (i) By allowing oracle content-aware augmentation, we can understand the upper-bound of different frameworks that rely upon augmentations. (ii) By creating a k-class classification task, we provide a more difficult task than existing binary datasets and enable the practitioners to control the difficulty of the task through k. (iii) Many graph datasets are relatively small compared to their image counterparts. However, large batch sizes are essential to the performance of many negative sample frameworks. Our benchmark allows for such frameworks to be evaluated in an appropriate setting.
> >
> > Next, we discuss several *natural extensions of our synthetic data generation process*. The current data generation process does not consider a dependency between the label and the type of style or content present in each sample. One simple modification is to encode such a dependency. For example, the type of motif (i.e. class) could also influence the structure of the background graph. Samples could have background graphs that are generated using different random graph generators (tree, Barbasi Albert, Power-Law, etc) where the probability of a given generator being used is dependent on the class. Another possibility is to introduce more complicated content information. While our current formulation uses the same motif over class samples, an alternative is to use variations of the same motif to determine the class. For example, if a 3x3 grid-motif corresponded to a particular class, then we can generate different sized grids (4x4, 2x2, 5x5, etc.) that are also from the same class. Here, the content is the concept of a grid-structure vs. identifying a particular grid. *This discussion will later be included in an updated version of the paper*. If necessary, we can also conduct additional experiments on such datasets but emphasize that the validity of our claims remains on simple datasets.
> >
> > + ***Concerns on proposing a better data augmentation or sampling method.***
> >
> > Our data generation process ensures that style and content are well-defined in each sample. Proposing a better data augmentation strategy or sampling method that can automatically preserve content while only perturbing the style for arbitrary datasets is arguably a grand challenge for contrastive learning. It is our intention that the community finds this work useful in building towards such an augmentation. However, we note that *our experiments using oracle context aware augmentations demonstrated that there is a limit to the gains that can be realized by improving augmentations alone.* To close the gap between supervised and unsupervised performance, the community must investigate other components of the unsupervised graph representation learning paradigm.
> >
> > We note that for practical datasets encoding domain knowledge on what information is important or irrelevant to a decision is one approach to designing better augmentations that respect this partition. For example, on molecular datasets, carbon rings, a common structure, are not very indicative of a given molecule’s function. This information can be treated as the style and augmentations may perturb these structures. Functional groups in a molecule are indicative of the overall function. Therefore, these groups can be treated as the content, and augmentations should not modify this information.
> > For datasets where domain knowledge is not readily available, i.e. common benchmarks, generic graph augmentations are often used. Our analysis of GGA is useful as it helps us understand the behavior of these popularly used augmentations. *To that end, our paper contains an empirical analysis that showcases the unreasonable effectiveness of untrained GNNs, analyzes the performance of commonly used augmentations and proposes a synthetic benchmark that highlights that even the best augmentation have limits.*

---

> > > ### Comment · Reviewer_UMWA · 2021-12-01
> > > **thanks to the rebuttal**
> > >
> > > i appreciate it greatly that the authors give the detailed responses. the motivation and idea is in general interesting while i will still keep my score since it still needs a major revision before satisfactory.

---

### Author Response · Authors · 2021-11-16
**Summary Response on Novelty and Signifiance**

> *We thank all the reviewers for their constructive feedback. In light of reviewer comments, we clarify our contributions and the significance of the work. The main paper has also been modified to reflect this clarification.*

+ Unsupervised graph representation learning pipelines contain many different components whose joint interactions are not well understood. For example, it is unclear how encoder architecture, data augmentation, and training paradigms should be selected for strong downstream performance on a particular task. Understanding the impact of different components can enable practitioners to determine if pre-training will indeed improve representation quality and appropriately select components for their settings. In this paper, we focus on understanding two such crucial components: choice of self-supervised learning paradigm (contrastive learning vs. reconstruction-based), and augmentation design.
+ Following the organization of our modified paper, we begin by conducting an empirical study on standard benchmark datasets to demonstrate that there is not a winning combination that can be readily recommended to practitioners. Specifically, we compare different learning paradigms (CL vs. reconstruction-based) against untrained GNNs.
+ Our work is the **first to implement and benchmark augmentation-augmented graph auto-encoders (AAGAE)** as a stable reconstruction-based approach for graph SSL.
+ Moreover, we conduct an **extensive study on the unreasonable effectiveness of untrained models** across different architectures and introduce sample invariance as an additional metric for evaluating graph SSL representations.
+ While learning invariance to context-aware augmentations is known to improve performance in computer vision, such augmentations are not readily available for standard graph datasets. Therefore, we **extend analytical tools from visual CL to characterize the behaviour of generic graph augmentations (GGA).** Our analysis uses GED between dataset samples, a fundamental property of the dataset, and the population augmentation graph (HaoChen et. al) to understand the behavior of GGA and its effects on GCL. To the best of our knowledge, we are the first to conduct such an analysis.
+ Lastly, we **introduce a novel, synthetic benchmark** to enable systematic evaluation of the potential gains to be obtained from context-aware augmentations. We **use a style vs. content perspective to understand the performance** of reconstruction vs. contrastive learning paradigms. We make a surprising finding that, even with oracle context-aware augmentations and sophisticated GNN architectures, we are unable to close the performance gap with respect to supervised learning. This is in stark contrast to the success of visual CL and suggests that other pipeline components must also be improved, if graph representation is to achieve similar success.

In summary, our work offers empirical and theoretial analysis and a useful benchmark that future works can use to evaluate the benefits of unsupervised graph representation learning methods.

---

### Author Response · Authors · 2021-11-16
**Summary Response on Restructuring PAG-based Analysis of Generic Graph Augmentations**

> *We have incorporated reviewer feedback to restructure and expand our analysis in Section 3 to be easier to follow and make our contributions more clear. Please see the main paper, and Appendix A.1 for the improved version. We summarize our updated analysis here.*

+ The discussion on relationship between GGA and GED has been expanded and now accounts for categorical node attributes when computing GED through a "replacement" operation. (Section A.1.1).
+ We add definitions for augmentation sets, overlapping samples, and invalid samples based on GED.
+ Remark A.1 demonstrates how to determine the size of augmentation sets.
+ Remark A.2 defines a conditions for the existence of overlapping samples using GED.
+ Claim A.1 computes a lower-bound on the number of overlapping samples and invalid samples given a GED distance matrix, augmentation strength, and a label vector.
+ By using the above definition and claims based that are not based on the PAG, our updated analysis is applicable to any method that uses GGA. We also identify a conceptual min-max optimization problem between the size of augmentation sets, and the number of overlapping or invalid samples. *This optimization highlights that the central assumption of GGA, that small changes to a graph will not alter its label, is incomplete. We must also consider the distance to other samples when determining an augmentation strength that preserves labels.*
+ Section A.1.3 introduces and constructs the Population Augmentation Graph. Its construction is now more straightforward because it follows directly from the preceding discussion and definitions.
+ Section A.1.4 formalizes and expands the discussion on GED its influence on PAG topology.
+ Claim A.2 demonstrates that an augmented and overlapping sample will have a higher degree in the PAG than if the same sample was not an overlapping sample, establishing a simple but clear relationship between GED and the structure of the PAG.
+ Claim A.3 discusses more generally how GED determines PAG structure. We show that edges only exist between samples belonging to the same augmentation set and overlapping edges (an edge between two overlapping nodes) have increased edge weights.
+ Claim A.4 expands our original analysis to include a discussion on the interplay between GED and the recoverability assumptions made by HaoChen et. al for their theoretical analysis. Specifically, it provides a lower bound on the error of classifying augmented samples (i.e. recoverability). If we assume that labels of invalid samples cannot be well recovered, then the classifier's error is dependent on how often such a sample occurs and this error cannot be reduced further. Popular GCL frameworks often borrow from successful visual CL frameworks. Therefore, it is important to recognize when assumptions from  visual frameworks do not hold for GCL.

---

### Author Response · Authors · 2021-11-29
**Gentle Reminder to Reviewers**

Dear Reviewers,

Since today is the last day in the discussion period, can you please take a look at our revised paper and rebuttal responses to let us know if we have adequately addressed your concerns? We are happy to address any further questions.

Thank you!

---

### Decision · Program_Chairs · 2022-01-20

**Decision:**

Reject

**Comment:**

In this paper, data augmentation for graph contrastive learning (GraphCL) is studied. Most reviewers agree that the problems addressed in this paper are interesting and important for unsupervised graph representation learning literature. However, many reviewers were not fully satisfied with the novelty and the claim of the main contribution of this paper, a theoretical analysis of the conditions under which data augmentation works in GraphCL, due to the lack of clear explanation and evidence. Unfortunately, no reviewer has suggested acceptance of this paper at this time.